

# Intercomparison of in-situ NDIR and column FTIR
# measurements of $CO_2$ at Jungfraujoch
**Michael F. Schibig[1], Emmanuel Mahieu[2], Stephan Henne[3], Bernard Lejeune [2],**
**Markus C. Leuenberger[1]**
[1] Climate and Environmental Physics, Physics Institute and Oeschger Centre for Climate
Change Research, University of Bern, Bern, Switzerland
[2] Institut d'Astrophysique et de Géophysique, Université de Liège, Liège, Belgique
[3] Empa, Swiss Federal Laboratories for Materials Testing and Research, Dübendorf,
Switzerland
Correspondence to: leuenberger@climate.unibe.ch
Keywords: $CO_2$, FTIR, NDIR, Jungfraujoch, intercomparison, $CO_2$ trend
**Abstract**
We compare two $CO_2$ time series measured at the High Alpine Research Station Jungfraujoch
(3580 m a.s.l., Switzerland) in the period from 2005 to 2013 with an in-situ surface
measurement system using a nondispersive infrared analyzer (NDIR) and a ground-based
remote sensing system using solar absorption Fourier Transform Infrared spectrometry
(FTIR). Although the two data sets show an absolute shift of about 13 ppm, the slopes of the
annual $CO_2$ increase are in good agreement within their uncertainties. They are $2.04 \pm 0.07$
ppm yr$^{-1}$ and $1.97 \pm 0.05$ ppm yr$^{-1}$ for the FTIR and the NDIR system, respectively. The
seasonality of the FTIR and the NDIR system is $4.46 \pm 1.11$ ppm and $10.10 \pm 0.73$ ppm,
respectively. The difference is caused by a dampening of the $CO_2$ signal with increasing
altitude due to mixing processes. While the minima of both data series occur in the middle of
August, the maxima of the two datasets differ by about ten weeks, the maximum of the FTIR
measurements is in middle of January, whereas the maximum of the NDIR measurements is
found at the end of March. Sensitivity analyses revealed that the air masses measured by the
NDIR system at the surface of Jungfraujoch are mainly influenced by central Europe, whereas
the air masses measured by the FTIR system in the column above Jungfraujoch are influenced
by regions as far west as the Caribbean and the United States.



The correlation between the hourly averaged $CO_2$ values of the NDIR system and the
individual FTIR $CO_2$ measurements is 0.820, which is very encouraging given the largely
different sampling volumes. Further correlation analyses showed, that the correlation is
mainly driven by the annual $CO_2$ increase and to a lesser degree by the seasonality. Both
systems are suitable to monitor the long-term $CO_2$ increase, because this signal is represented
in the whole atmosphere due to mixing.
**1    Introduction**
$CO_2$ is the most important anthropogenic greenhouse gas, with a large contribution to the
greenhouse effect (Arrhenius, 1896) and an additional radiative forcing of the atmosphere
currently evaluated at 1.68 $Wm^{-2}$ (IPCC, 2013). The strength of the forcing is depending on
its atmospheric mole fraction which is ruled by the processes of the carbon cycle as well as by
anthropogenic $CO_2$ emissions from fossil fuel combustion and land use change. The major
reservoirs of the carbon cycle besides the lithosphere are the soils, the ocean, the biosphere
and the atmosphere, where the latter is also acting as the main link between the biosphere and
the ocean. The processes coupling the biosphere with the atmosphere are photosynthesis,
where $CO_2$ is used by plants to convert solar energy into chemical energy by producing
carbohydrates from $CO_2$ and $H_2O$, and respiration, the decomposition of biogenic
carbohydrates back into $CO_2$, $H_2O$ and energy, where $CO_2$ is released back to the atmosphere.
The linking process between the atmosphere and the ocean is dissolution of $CO_2$ in oceanic
water, where it is subsequently chemically bound to bicarbonate and carbonate and therefore
removed from the carbon cycle on a longer timescale (Broecker and Peng, 1982;Feely et al.,
2004;Heinze et al., 1991;Sillén, 1966). The solution of $CO_2$ in water is depending on the
partial pressures of $CO_2$ in the atmosphere and the ocean, if the atmospheric partial pressure
of $CO_2$ above sea water is greater than the oceanic partial pressure of $CO_2$, $CO_2$ is taken up by
the seawater and vice versa. Other factors as e.g. salinity, temperature etc. affect the solubility
of $CO_2$ in seawater additionally (Bohr, 1899;Takahashi et al., 2009). Photosynthesis and
respiration, on the other hand, are mainly driven by climatic conditions of the environment. In
the northern hemisphere, especially in the extratropics with distinct seasons, the dominating
process in late spring, summer and fall is photosynthesis and thereby the uptake of $CO_2$ from
the atmosphere. In autumn respiration and with it the release of $CO_2$ from the biosphere into
the atmosphere starts to take over and is the ruling process in winter until spring when





photosynthesis becomes the dominating process again. Due to these alternating processes, the
$CO_2$ mole fraction in the atmosphere shows a seasonal cycle with its maximum generally in
early spring and its minimum in fall (Halloran, 2012;Keeling et al., 1976;Keeling et al.,
2001;Machida et al., 2002). A further component in the change of atmospheric $CO_2$ mole
fraction is $CO_2$ release due to fossil fuel combustion (Karl and Trenberth, 2003;Revelle and
Suess, 1957;Tans et al., 1990). Nowadays, roughly half of the anthropogenically produced
$CO_2$ ends up in the oceans and the biosphere, whereas the other half is accumulating in the
atmosphere and leads to a more or less steady increase of the atmospheric $CO_2$ mole fraction
(Bender et al., 2005;Le Quéré et al., 2013;Sabine et al., 2004). Measuring the atmosphere's
$CO_2$ mole fraction on the long-term is therefore important to understand the sources and sinks
of the carbon cycle and the annual $CO_2$ increase due to fossil fuel combustion and land use
change. To measure the evolution of $CO_2$ in the atmosphere on a global scale satellite remote
sensing methods can be used as e.g. OCO-2 (Crisp et al., 2004, Pollock et al., 2010,
Thompson et al., 2012) or GOSAT (Chevallier et al., 2009, Yokota et al., 2009) but they are
limited by e.g. cloud cover, temporal coverage due to the orbit, coarse resolution etc. An
intercomparison between GOSAT and several TCCON (Total Carbon Column Observation
Network) stations showed a mean difference for daily averages of -0.34 ± 1.37 ppm
(Heymann et al., 2015). Ground based measurement systems on the other hand have a high
temporal resolution and provide very accurate data, which can be used to validate satellite
data (Buchwitz et al., 2006; Butz et al., 2011;Dils et al., 2006; Morino et al., 2011;Wunch et
al., 2011) or as model input (Chevallier et al., 2010), but surface observations have often a
limited representativeness and are often influenced by nearby processes and hence, not
representative for larger areas. Also the influence of the biosphere or anthropogenic pollution
can be a serious issue and make it very challenging to measure background air. Therefore, to
measure global $CO_2$ trends the sampling site should be at a very remote place like e.g. Mace
Head Station (Bousquet et al., 1996;Messager et al., 2008) on the western coast of Ireland or
the flask sampling network in the Pacific of NOAA (Komhyr et al., 1985;Trolier et al., 1996).
Another possibility is to measure in the free troposphere e.g. with airplanes as done in the
CARIBIC project (Brenninkmeijer et al., 2007) or the CONTRAIL project (Machida et al.,
2008) or at high altitudes which are mostly in the free troposphere as e.g. Mauna Loa
(Keeling et al., 1976;Keeling et al., 1995;Pales and Keeling, 1965;Thoning et al., 1989). The
High Alpine Research Station Jungfraujoch (JFJ) with its altitude of 3580 m a.s.l. (Sphinx





Observatory) and position mostly above the planetary boundary (Henne et al., 2010) is
therefore a very suitable spot to conduct ground based $CO_2$ background measurements.
The University of Liège (Belgium) has been measuring infrared radiation at JFJ since the
1950s and started regular FTIR (Fourier Transform InfraRed) measurements in 1984. The
Climate and Environmental Physics Division (KUP) of the University of Bern started
measuring $CO_2$ and $\delta O_2/N_2$ in 2000 by a flask sampling program and since the end of 2004,
$CO_2$ and $O_2$ have been additionally measured with a continuously operating system of a NDIR
instrument and a paramagnetic cell. In this study we compared the FTIR and the NDIR data
set to see if the two complementary measurement techniques are catching the same trends,
seasonalities and variations in atmospheric $CO_2$ mole fraction at and above Jungfraujoch.
**2    Methods**
**2.1    Measurement site**
The High Altitude Research Station Jungfraujoch (JFJ) is located 7°59'02'' E, 46°32'53'' N
at the northern margin of the Swiss Alps. The Jungfraujoch is a mountain saddle between the
Mönch (4099 m a.s.l.) and Jungfrau (4158 m a.s.l.) summits at a height of 3580 m a.s.l.
(Sphinx Observatory) and is accessible year-round by train. Because of the high elevation, the
station is usually above the planetary boundary layer (PBL) and therefore mainly receives air
from the free troposphere which is why it was classified as "mostly remote" by Henne et al.
(2010). Nevertheless, the station can be influenced by polluted air during specific events such
as frontal passages and Föhn (Uglietti et al., 2011;Zellweger et al., 2003) or thermal uplift of
polluted air from the surrounding valleys on fair weather days (Baltensperger et al., 1997;
Henne et al., 2005;Zellweger et al., 2000). Because of the high elevation, the accessibility and
the good infrastructure, the JFJ is an ideal location for in-situ measurements of atmospheric
background air from continental Europe (Baltensperger et al., 1997;Henne et al.,
2010;Zellweger et al., 2003). JFJ is also one of the currently 29 core sites of the WMO GAW
(Global Atmospheric Watch) programme.
**2.2    In-situ NDIR measurements at Jungfraujoch**
The KUP $CO_2$ measurements are based on a combined system to monitor $CO_2$ and $O_2$
changes in the atmosphere. The ambient air is entering through a strongly ventilated (600





$m^3\ h^{-1}$) common inlet on the observatory's roof to a manifold, which serves many trace gas
analyzers, where an aliquot of it is drawn to the KUP system. The air is cryogenically dried to
a dew point of -90 °C (FC-100D21, FTS systems, USA). Temperature as well as pressure is
stabilized to avoid influences caused by ambient air density fluctuations. This allows the
determination of $CO_2$ by a NDIR spectrometer (Maihak S710) with a frequency of 1 Hz and
$O_2$ by a paramagnetic cell under highly controlled conditions. Measurements are done in a
cyclic sequence of 18 hours with each gas measured for 6 minutes with only the last 115
seconds of a six minute period used for mole fraction determination, to allow for signal
stabilization after changing the sample source. At the beginning of each 18-hour sequence, the
system is calibrated with two reference gases (high and low span). A working gas is measured
between two ambient air measurements to correct for short term variations. All measurements
ending in a particular hour are used for the calculation of hourly mean $CO_2$ observations,
which in our case includes therefore 6 ambient observation values per hour. Cylinder
measurements with a known mole fraction showed a precision better than 0.04 ppm for 1 hour
analysis. The $CO_2$ values are reported on the WMO X2007 scale. A multi-annual
intercomparison between the NDIR system and a cavity ring-down spectroscope at JFJ
showed a very good agreement of the $CO_2$ measurements (Schibig et al., 2015).
**2.3    Column FTIR measurements at Jungfraujoch**
The University of Liège has been recording atmospheric solar spectra at JFJ since the early
1950s. The current FTIR instrument is a commercially available Bruker IFS-120 HR with a
resolution of up to 0.001 $cm^{-1}$ (Mahieu et al., 1997). It features interchangeable detectors, a
KBr beam-splitter and dedicated optical filters, which altogether give the possibility to cover
the 1 to 14 µm spectral range (Zander et al., 2008). Here gases such as $CO_2$, $CH_4$ and $H_2O$
show numerous absorption lines documenting contributions to the greenhouse effect. These
spectra also contain information about the abundance of many additional absorbing gas
species in the path between the instrument and the sun, essentially present either in the
troposphere or in the stratosphere. The $CO_2$ data set used here has been derived from the
reference total column time series produced within the framework of the NDACC monitoring
program (Network for the Detection of Atmospheric Composition Change; see
http://www.ndacc.org), presented previously in e.g. Zander et al. (2008; see Figure 6). The
uncertainty on the main $CO_2$ line strength is estimated at 2 to less than 5% in the HITRAN
compilation (Rothman et al., 2005), leading to a systematic error on the retrieved total column





of the same magnitude. In the meantime, the data set has been consistently updated, still using the SFIT-1 algorithm (version 1.09c) and a single microwindow spanning the 2024.3 – 2024.7 $cm^{-1}$ spectral interval, whose main spectral line is coming from $^{13}CO_2$. The single $CO_2$ a priori vertical distribution used in all retrievals is characterized by a constant mixing ratio of 338 ppm from the surface up to the tropopause, then slightly decreasing to stabilize at 330 ppm at 20 km and above. A simple scaling retrieval is performed, and the mixing ratio derived for the troposphere is used in the present comparisons. Note that the representativeness of this unique profile is not optimal for all seasons and may lead to an underestimation of the seasonal amplitude (see Fig. 1 in Barthlott et al., 2015), because of a non-optimum vertical sensitivity of the FTIR retrieval. Indeed, typical values of the total column averaging kernel – indicative of the fraction of information coming from retrieval rather than from the a priori (e.g. Vigouroux et al., 2015) – are in the 0.5 – 1 range between the ground and 10 km altitude, in line with Fig. 4 of Barthlott et al. (2015).

## 2.4    Data processing

The NDIR data set is much more influenced by near ground processes like thermal uplift of PBL air from the surrounding valleys, advection of PBL air by synoptic events etc. than the FTIR and shows therefore a higher variability. Additionally, because of the large volume of the column sampled by the FTIR above JFJ the $CO_2$ mole fraction measured by the FTIR is averaged and the data set is far less sensitive to local events than the in-situ NDIR measurements. The FTIR needs a cloudless sky to be able to measure, whereas the NDIR system is measuring under all conditions, which can lead to very high $CO_2$ mole fractions during e.g. Föhn events, when the sky is cloudy and polluted air from the heavily industrialized Po basin (Northern Italy) is advected to JFJ. Therefore, only measurements of background air should be taken into account to compare the two data sets properly.

### 2.4.1    Filtering, trend and seasonality calculation

The background data were selected using a statistical approach. A cubic spline was fitted to both datasets individually, the standard deviation of the residuals was calculated and all points beyond 2.7 σ were flagged as outliers. This process was repeated in both data sets until convergence. The threshold of 2.7 σ was chosen because in normally distributed data more than 99 % of the total data points would be included for further calculations and only the most obvious outliers (less than 1 %) would be rejected.



The $CO_2$ mole fraction is dominated by two major processes. One is the linear increase due to
fossil fuel combustion (trend) and one is the annual in- and decrease due to respiration and
photosynthesis (seasonality). The trend was calculated for both datasets individually with a
Monte Carlo approach.
For the trend calculation we intentionally used the datasets including seasonal signals because
it leads to realistic trend error estimates compared to deseasonalized datasets, which in our
view tend to underestimate the error. The datasets were split in two subsets, where each of the
subsets spanned over n - 0.5 phases (in this study n equals 9 years) to prevent a bias in the
trend calculation due to the seasonal cycle. The first subsets start in January 2005, the second
subsets start in July 2005. In each subset about 2 % (a higher number does improve the result)
of the points were selected randomly and the linear trend was calculated. This was repeated
500 times with each subset and the averages of these linear trends were taken as the slopes of
the datasets.
To calculate the seasonality, the two datasets were detrended and monthly averages were
formed, from which the seasonality was calculated as the difference between the highest and
the lowest value.
**2.4.2   Correlation analysis**
Because of the different time resolutions for in-situ and FTIR measurements we selected
those in-situ measurements (six minute and hourly NDIR averages) that are closest (± 30 min)
to the FTIR values for correlation analysis.
Since the differences between both correlation analyses were negligible (see results section),
it was decided to continue with the hourly averages of the NDIR dataset only, which is the
common output of the NDIR database.
The FTIR's sample volume is much bigger than the NDIR system's and because of
transportation processes there's a possibility of mixing processes. To check, a moving average
of the NDIR data with increasing width was calculated to see if the correlation is enhanced
with expanding width (from 0 to ± 600 h).
Furthermore, the column measurements were retrieved for the layer between 3.58 km (altitude
of the Sphinx Observatory) to the top atmosphere (set to 100 km in the retrieval scheme)
whereas the NDIR system is measuring at the lower boundary of the FTIR's sampling
column, therefore it is possible that a time shift in the measured $CO_2$ mole fractions due to
advection, uplift of air parcels etc. occurs. To check whether a systematic time shift exists





between the two datasets, the NDIR measurements were shifted relative to the FTIR data
from -60 to +60 days (corresponding to -1440 h to +1440 h) in hourly steps and again the
correlation of the two data sets was calculated. If there is a systematic time shift, the deviation
should be indicated by increased correlation values.
**2.5    FLEXPART model runs**
From 2009 to 2011, backward Lagrangian particle dispersion model simulations were
performed with FLEXPART (Stohl, et al. 2005) to simulate the transport towards JFJ and
estimate surface source sensitivities (footprints) of the sampled air masses. To account for the
complex flow in the Alpine area, a regional scale version of the model driven by operational
output from the regional scale numerical weather prediction model COSMO as produced by
MeteoSwiss was used (Henne et al., 2015, Oney et al., 2015) . Since COSMO is a limited area
model, the transport of particles leaving the domain was further simulated in the global scale
version of FLEXPART (Stohl et al., 2005) driven by operational analysis fields of the
European Centre for Medium Range Weather Forecast (ECMWF). In the Alpine area,
COSMO input data had a horizontal resolution of approximately 2 km x 2 km, in Western
Europe 7 km x 7 km. Of the 1214 FTIR measurements in this period, footprints were
available for 766. The model simulated footprints of the surface in-situ observations and five
partial columns above JFJ reaching from 3365-4226 m a.s.l., 4226-4912 m a.s.l., 4912-5629
m a.s.l., 5629-6386 m a.s.l. and 6386-7184 m a.s.l. The lower boundary is below JFJ in order
to account for smoothed model topography. Particles released at and above JFJ were followed
10 days backward in time to calculate source sensitivities. Source sensitivities were evaluated
on regular longitude/latitude grids. The resolution was 0.5° x 0.5° globally, 0.2° x 0.2° over
Europe and an even higher resolution of 0.1° x 0.1° was used in the Alpine area. The
footprints of the individual measurements of each partial column were averaged to monthly
means to get information about the origin of the air masses in the according month (Henne,
2014;Henne et al., 2013).
**3    Results**
Because of the different measurement techniques, the number of data points in the two
datasets is different. In the period 2005 to 2013 the NDIR dataset contains 68477 hourly
averages from which about 5 % were omitted as pollution or depletion events resulting from



PBL influence as estimated by the filtering (Figure 1). In the same period, the FTIR dataset
shows 3068 measurements of which about 5 % were rejected as  pollution and depletion
events, too (Figure 2). For all further calculations, only the filtered datasets were used.
The average of the detrended and deseasonalized NDIR data before and after filtering was
$0.00 \pm 2.65$ ppm and $0.00 \pm 1.84$ ppm (Figure 3 A), the average of the FTIR data was $0.01 \pm$
$2.61$ ppm and $0.01 \pm 2.16$ ppm, respectively (Figure 3 B).
With a Monte Carlo algorithm, the values of the annual change of the $CO_2$ mole fraction of
the two datasets were calculated. Despite the shift between the two datasets of roughly 13
ppm (i.e. about 3%, in line with the systematic uncertainty affecting the FTIR measurement;
see section 2.3) and the different measurement techniques the annual $CO_2$ increase is quite
similar. The FTIR slope is $2.04 \pm 0.07$ ppm yr$^{-1}$ and the NDIR dataset shows a slope of $1.97 \pm$
$0.05$ ppm yr$^{-1}$, so they are equal within their uncertainties (Figure 4).
By detrending the datasets with the derived slopes, the seasonality can be calculated. The
column dataset shows a seasonality of $4.46 \pm 1.11$ ppm whereas the in-situ measurements at
the Sphinx Observatory show a seasonality roughly twice as big, namely $10.10 \pm 0.73$ ppm. To
find the moment of the average minima and maxima, a two harmonic fit function was applied
to the detrended datasets. The minima of the FTIR and NDIR datasets are both in the middle
of August, but the maxima are roughly ten weeks apart. The maximum of the NDIR datasets
occurs at the end of March, whereas seasonality of the FTIR dataset already reaches its
maximum in the middle of January (Figure 5).
The footprints of August, January and March, when the extrema of the seasonal cycle
occurred,  as calculated with FLEXPART show that the in-situ observation at Jungfraujoch is
mainly receiving air masses that are  influenced by Central Europe, and to a lesser degree by
the Mediterranean area and the northern Atlantic (Figure 6, Figure 7 and Figure 8).
With increasing altitude, the footprints of the sub-columns indicate, that the measured air
masses become more sensitive to regions as far west as e.g. the Caribbean and the United
States and that the influence from the European continent and northern regions higher than
50°N is decreasing (Figure 6, Figure 7 and Figure 8).
To estimate the relationship between the FTIR and NDIR measurements the correlation was
calculated. The FTIR measurements take normally about 10 min and are done whenever
possible. Therefore the FTIR data is reported exactly at the measuring time. The NDIR on the
other hand is measuring non-stop, but only 115 s of six-minute intervals (see methods) are
used to calculate a data point and the six-minute data is normally averaged to hourly averages.




Therefore we first checked whether the high resolution data are necessary or hourly data is
good enough. To do so, to each FTIR data point the nearest high resolution and hourly
averaged NDIR values were assigned. An additional condition was that the NDIR value must
not be further apart than ± 30 min, otherwise no NDIR data point was set, which was the case
in about 10 % of the FTIR data points. The correlation between the FTIR and the high
resolution NDIR $CO_2$ measurements and between the FTIR and the hourly averages were
calculated to be 0.819 and 0.820, so the differences between the two regression values are
negligible. To examine the relationship between the FTIR and the NDIR measurements
further, the seasonality of the two datasets was eliminated which gave almost the same
correlation of 0.824 (0.838 with the high resolution data). In the next step only the trend was
subtracted and the remaining seasonalities were compared, which lead to a much smaller
correlation of 0.460 (0.461 with the high resolution data). In a final step, the trend as well as
the seasonality was removed, which resulted in a correlation of 0.071 (0.084 high resolution
data vs. FTIR). Since correlations between the FTIR data and the NDIR's high resolution and
the hourly data were almost the same, only the hourly data was considered for further
calculations (Figure 9).
As mentioned above, the column measurements represent the whole vertical distribution
above Jungfraujoch whereas the NDIR system is measuring at the base of the FTIR's
sampling column. Therefore, the two records might be time-delayed due to advection, uplift
of air parcels etc. To check for a potential time lag, the NDIR measurements were shifted
relative to the FTIR data from -1440 to +1440 hours in hourly steps.
The correlations between the NDIR and FTIR datasets and between the deseasonalized NDIR
and FTIR datasets show a peak region at a time shift from -10 h to 60 h with the highest
correlation being 0.830 and 0.836 respectively (Figure 10 A, Figure 10 B). The correlation
between the datasets is decreasing before and after this range, in the deseasonalized datasets
the correlation stays more or less stable. The correlation between the two trend corrected
datasets shows a plateau of enhanced correlation values from -50 h to 200 h time shift with a
maximum correlation of 0.495 at a time shift of 165 h, at lower and higher time shifts, the
correlation is decreasing (Figure 10 C). The correlation of the detrended and deseasonalized
datasets shows no distinct pattern and is oscillating around 0 (Figure 10 D).
Since the air volume measured by the FTIR is much bigger than the NDIR system's volume,
vertical mixing and transport processes can occur and thereby changing the $CO_2$ mole fraction
in the measured air parcels. Therefore moving averages with increasing widths (up to ± 600 h)



were calculated from the NDIR data and the obtained averaged NDIR values were correlated
with the filtered FTIR dataset. Changing the width of the moving average doesn't have a
strong influence on the correlation between the two filtered datasets, because the increasing
width of the moving average just smooths the dataset. The correlation remains at about 0.85
(Figure 11 A), with a very small increase of the correlation at the beginning, most probably
due to the above mentioned smoothing effect. The same is true for the correlation between the
deseasonalized datasets. They show high correlation of about 0.84 over the whole range of
widths, with a slight increase at the beginning, which is not significant (Figure 11 B). By
detrending the datasets, the correlation is increasing with the width of the moving average and
shows a plateau of higher correlation of about 0.5 at a width 150 to 600 h from where on it is
decreasing again (Figure 11 C). However, the changes in the correlation within the range of
150 h to 600 h are very small. The detrended and deseasonalized datasets show a very low
correlation and the improvement of the correlation due to the changing width of the moving
average is negligible. Over all, the improvement of the correlations due to the changing width
of the moving average is very small (Figure 11 D).
Finally both, the time shift and the width of the moving average were varied about ± 1440 h
and ± 600 h, to see with which combination of time shift and width the best correlation can be
reached. They all show a ridge of higher correlation at a time shift around zero which is
broadening with increasing width of the moving average, except for the data without slope
and seasonality, which have a low correlation anyway (Figure 12). The increasing width of
the moving average leads to a small improvement of the correlations in the beginning,
however over all it doesn't seem to have a strong influence on the correlations. The time shift
on the other hand has an influence on correlation between the complete filtered datasets and
even more on the correlation of the detrended datasets. In the correlation of the
deseasonalized datasets, the influence of the time shift is very limited except for the small
ridge of slightly enhanced correlations around zero time shift as mentioned above.

## 4   Discussion

The filtered FTIR and NDIR datasets show a very similar increase in the $CO_2$ mole fraction of
ambient air, despite the two totally different measurement principles. The calculated annual
$CO_2$ trends of the FTIR and NDIR datasets are $2.04 \pm 0.07$ ppm $yr^{-1}$ and $1.97 \pm 0.05$ ppm $yr^{-1}$
respectively (Figure 4) and are in good agreement with flask measurements done at JFJ with a





slope of 1.85 ppm yr$^{-1}$ (van der Laan-Luijkx et al., 2013) and other remote stations in the
northern hemisphere; for example Mauna Loa with 2.05 ppm yr$^{-1}$ (Tans and Keeling, 2014) or
Alert with 1.85 ppm yr$^{-1}$ (Keeling et al., 2001). Also the NDIR dataset's average seasonality
of $10.10 \pm 0.73$ ppm is in good agreement with the seasonality of these flask measurements,
which were $10.54 \pm 0.18$ ppm in the period 2007 to 2011 (van der Laan-Luijkx et al., 2013)
and is roughly double the FTIR's average seasonality of $4.46 \pm 1.11$ ppm (Figure 5). The
lower seasonality of the FTIR dataset can be explained by the fact that the NDIR system is
measuring $CO_2$ mole fractions at the Sphinx Observatory, which is most of the time above the
PBL (Henne et al., 2010) but still closer to the ground than the FTIR measurements.
Therefore the signal of the biosphere is stronger than in the column, where it is attenuated by
vertical mixing and transport processes of the atmosphere with increasing height. Also the
fixed a priori vertical $CO_2$ profile may contribute partly to the lower seasonality of the FTIR
measurements. The shape of the profile used to retrieve the $CO_2$ data doesn't reproduce the
changes due to seasonality and is therefore not always the optimum. By using a seasonally
varying a priori retrieval the seasonality might be slightly higher because the amplitude of
$CO_2$ is better retrieved (Barthlott et al., 2015). Furthermore, in the tropopause and the lower
stratosphere, the phase of the $CO_2$ seasonality is shifted by several months (Bönisch et al.,
2008;Gurk et al., 2008;Bönisch et al., 2009). However, this has only a minor influence on the
observed dampening of the amplitude of the FTIR seasonality compared to the vertical
mixing, since the stratosphere contains only about 10 % of the abundance of atmospheric air
molecules.
It is not easy to define the seasonal minimum and maximum in the FTIR dataset because they
are not very clearly pronounced. By fitting a two harmonic function the minimum was found
to be in the middle of August, the maximum in the middle of January. While the minimum of
the NDIR dataset is around the same time, the maximum of the FTIR dataset occurs roughly
ten weeks earlier than the maxima of the NDIR dataset (Figure 5). The timing of the minima
of both datasets and the maximum of the NDIR dataset coincide quite well with net land-
atmosphere carbon flux changes from negative to positive values and vice versa (Zeng et al.,
2014). Therefore an alternative explanation is needed for the early maximum of the FTIR
dataset. Sensitivity analyses revealed that the upper tropospheric air originates from lower
latitudes than the in-situ air measured by the NDIR. Therefore the air measured by the FTIR
is partially decoupled from the increasing $CO_2$ values of the winter-time northern hemisphere.
Furthermore, the decoupling might be amplified by the weak overturn of tropospheric air in



winter. Towards spring, the tropospheric overturn speeds up again which results in
synchronous $CO_2$ minima for both datasets in August. Similar studies investigating CO at JFJ
also showed that JFJ is not only sensitive to Central Europe but also to regions as far west as
for example North America, the Pacific or even Asia and that the influence of these regions is
getting stronger with increasing height (Dils et al., 2011;Pfister et al., 2004;Zellweger et al.,
2009). Additionally, the assumption of a fixed a priori $CO_2$ vertical distribution to retrieve the
column integrated $CO_2$ concentration from the FTIR dataset may contribute partially to the
observed shift of ten weeks in the NDIR and FTIR maxima, because it is representing the
distribution in winter/spring inadequately.
Another hint that the two systems are not measuring the same air parcels can be found in
correlation analyses. After omitting outliers, which are mostly caused by synoptic events,
thermal uplift of polluted air from surrounding valleys, or other local to regional transport
events, the correlation of the two datasets is as large as 0.820, which is quite encouraging
considering the different nature of the measurements. By excluding the seasonality from both
datasets, the correlation stays almost the same, namely 0.824 but drops to 0.460 if the
seasonality is included but the annual $CO_2$ increase is subtracted. The comparison of the two
$CO_2$ datasets with the annual $CO_2$ increase and the seasonality subtracted showed a very low
correlation of 0.071, which is negligible (Figure 9). Because of possible delays and mixing
effects of the $CO_2$ signal, the time shift as well as the width of the moving average calculated
on the hourly values of the NDIR $CO_2$ values was varied between ± 1440 h and up to ± 600 h,
respectively. Shifting the NDIR time relative to the FTIR measurement time creates a ridge of
higher correlations around 0 h time shift with a slight tendency towards positive values
(Figure 12 A). This ridge-like form is clearly pronounced in the correlation plot between the
complete filtered FTIR and NDIR datasets and even more in the datasets without slope
(Figure 12 C) than in the correlation of the datasets without seasonality (Figure 12 B). There
it is very small and the correlation is high across the whole time shift and averaging width.
The constantly high correlation for deseasonalized datasets is due to both datasets containing
mostly background air whose $CO_2$ mole fraction changes are mainly driven by the annual $CO_2$
increase and by the seasonality of the $CO_2$ signal. Since the larger of the two, the seasonality,
is subtracted the high correlation is mainly driven by the slope which was calculated to be the
same within uncertainties and stays more or less constant over the examined period.
Therefore, the time shift has almost no influence. The remaining fluctuations in the $CO_2$ mole
fractions with higher frequencies than the seasonality seem to play a minor role, because



they're almost not visible in the comparison of the datasets without seasonality except for the
small ridge (Figure 12 B), or there's no correlation at all, as in the comparison of the two
datasets without slope and seasonality (Figure 12 D). This is indicating that the two
measurement systems are not measuring the same air parcels, even not with a certain delay, or
that the $CO_2$ signal of the NDIR system which is measured at the lower end of the FTIR
column becomes diluted beyond recognition for FTIR by the air mixing processes. The
positive effect of the increasing width of the moving average on the correlation is strongest,
but still very low, around the first 100 h. Afterwards its main effect is broadening the ridge of
the slightly enhanced correlations. The reason for the broadening effect of the increasing
width is its smoothing effect on the NDIR values. With increasing width, the influence of a
specific NDIR point on the correlation becomes smaller and the NDIR dataset evolves into a
smooth sine like curve with decreasing amplitudes, similar to the FTIR dataset, where this
form is caused by the higher sampling volume and the dampening due to mixing processes in
the atmosphere. However, the small influence of the moving average's width on the
correlation means that the correlation of the in-situ and the column measurement is mainly
influenced by the slope and the seasonality. Short term fluctuations play a minor role mainly
because either their $CO_2$ signal is dampened too much to be seen in the column measurement
or it is not measured at all as e.g. diurnal cycles because of the applied measurement methods.
**5  Conclusions**
Two datasets of $CO_2$ measurements at the High Altitude Research Station Jungfraujoch in the
period 2005 to 2013 were compared. The FTIR system is measuring the attenuation of solar
light at different wavelengths caused by molecules of light absorbing gas species in the
column between the Sphinx Observatory and the sun. From the obtained spectra, with the
knowledge of $CO_2$ specific extinction bands and the pressure distribution along the path of the
light, it is possible to calculate the $CO_2$ mole fraction in the column. The NDIR system is
measuring the $CO_2$ mole fraction of ambient air at the Sphinx Observatory which corresponds
to the lower boundary of the FTIR measurements. The two datasets were filtered with a
statistical approach to exclude $CO_2$ measurements which were influenced by recent transport
from the planetary boundary layer. The filtering caused a loss of about 5 % in both, the NDIR
and the FTIR dataset.
The annual $CO_2$ increase of the two datasets was calculated with a Monte Carlo approach.
Despite an average offset of 13 ppm between the two datasets, which is within the systematic
uncertainty affecting the FTIR measurement, the slopes were in good agreement, namely 2.04
$\pm$ 0.07 ppm yr$^{-1}$ in the FTIR measurements and 1.97 $\pm$ 0.05 ppm yr$^{-1}$ in the NDIR dataset. The
seasonality of the $CO_2$ signal of the NDIR and the FTIR system is 10.10 $\pm$ 0.73 ppm and 4.46
$\pm$ 1.11 ppm, respectively. The difference is caused by a dampening of the $CO_2$ signal with
increasing altitude due to mixing processes. While the minima of the two datasets both occur
in the simultaneously, the maxima of the FTIR dataset was found ten weeks earlier than the
NDIR maxima.
The difference in the occurrence of the minima is most probably caused by the different
transport history of the air masses measured at JFJ and in the column above JFJ. In January,
the in-situ system is measuring air from central Europe and the Mediterranean, whereas the
air masses of the column measurements are more affected by the subtropic Northern Atlantic.
With the onset of spring in Europe, the photosynthetic activity is increasing and the $CO_2$ mole
fraction of air measured by the in-situ system starts to decrease at the end of March. The two
filtered datasets as well as the two deseasonalized datasets show a high correlation, whereas
the correlation between the two detrended datasets is only mediocre and inexistent in the
between the two detrended and deseasonalized datasets. Neither shifting the time of the NDIR
measurements relative to the FTIR measurements nor increasing the width of the moving
average did increase the correlation between the two datasets significantly. The enhanced
correlation values around a time shift of zero indicates that (i) there isn't a systematic time
shift apparent and that (ii) the correlation between the two datasets is mainly driven by the
annual $CO_2$ increase and to a lesser degree by the seasonality. Therefore both measurement
systems are suitable to measure the annual $CO_2$ increase, because this signal is well mixed
within the atmosphere. Short term variations as the seasonality or daily variations are less or
not comparable, because (a) the transport history of the air parcels measured is different, (b)
the signal is mixed beyond recognition or (c) since the FTIR retrievals has little vertical
sensitivity the measured column signal contains mixed information from the troposphere and
the stratosphere.
**Acknowledgements**



This work was financially supported by the Swiss National Science Foundation (SNF-Project
200020_134641) and the Federal Office of Meteorology and Climatology MeteoSwiss in the
framework of Swiss GCOS. We like to thank the International Foundation High Altitude
Research Stations Jungfraujoch and Gornergrat (HFSJG), especially the custodians Martin
Fischer, Felix Seiler and Urs Otz for changing the calibration gases cylinders of the NDIR
system and other maintenance work. Additionally the authors like to thank Hanspeter Moret
and Peter Nyfeler for his precious work and help in maintaining and repairing the systems in
the Laboratory in Bern and also at Jungfraujoch. The Belgian contribution to the present work
was mainly supported by the Belgian Science Policy Office (BELSPO) and the Fonds de la
Recherche Scientifique – FNRS, both in Brussels. FLEXPART simulations were carried out
in the framework of EC FP7 project NORS (grant agreement N° 284421).  Additional support
was provided by MeteoSwiss (GAW-CH) and the Fédération Wallonie Bruxelles. We are
grateful to the many colleagues and collaborators who have contributed to FTIR data
acquisition.





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

Preliminary validation of column-averaged volume mixing ratios of carbon dioxide and
methane retrieved from GOSAT short-wavelength infrared spectra, Atmos. Meas. Tech., 4,
1061-1076, 10.5194/amt-4-1061-2011, 2011.
Oney, B., Henne, S., Gruber, N., Leuenberger, M., Bamberger, I., Eugster, W., and Brunner,
D.: The CarboCount CH sites: characterization of a dense greenhouse gas observation
network, Atmos. Chem. Phys. Discuss., 15, 12911-12956, 10.5194/acpd-15-12911-2015,
18  2015.

Pales, J. C., and Keeling, C. D.: The concentration of atmospheric carbon dioxide in Hawaii,
Journal of Geophysical Research, 70, 6053-6076, 10.1029/JZ070i024p06053, 1965.
Pfister, G., Pétron, G., Emmons, L. K., Gille, J. C., Edwards, D. P., Lamarque, J. F., Attie, J.
L., Granier, C., and Novelli, P. C.: Evaluation of CO simulations and the analysis of the CO
budget for Europe, Journal of Geophysical Research: Atmospheres, 109, D19304,
10.1029/2004JD004691, 2004.
Pollock, R., Haring, R. E., Holden, J. R., Johnson, D. L., Kapitanoff, A., Mohlman, D.,
Phillips, C., Randall, D., Rechsteiner, D., Rivera, J., Rodriguez, J. I., Schwochert, M. A., and
Sutin, B. M.: The Orbiting Carbon Observatory nstrument: performance of the OCO
instrument and plans for the OCO-2 instrument, 2010, 78260W-78260W-78213, 2010.
Rothman, L. S., Jacquemart, D., Barbe, A., Chris Benner, D., Birk, M., Brown, L. R., Carleer,
M. R., Chackerian, C., Chance, K., Coudert, L. H., Dana, V., Devi, V. M., Flaud, J.-M.,
Gamache, R. R., Goldman, A., Hartmann, J.-M., Jucks, K. W., Maki, A. G., Mandin, J.-Y.,





Massie, S. T., Orphal, J., Perrin, A., Rinsland, C. P., Smith, M. A. H., Tennyson, J.,
Tolchenov, R. N., Toth, R. A., Vander Auwera, J., Varanasi, P. and Wagner, G.: The
HITRAN 2004 molecular spectroscopic database, Journal of Quantitative Spectroscopy and
Radiative Transfer, 96(2), 139–204, doi:10.1016/j.jqsrt.2004.10.008, 2005.
Revelle, R., and Suess, H. E.: Carbon Dioxide Exchange Between Atmosphere and Ocean and
the Question of an Increase of Atmospheric $CO_2$ during the Past Decades, Tellus, 9, 18-27,
10.1111/j.2153-3490.1957.tb01849.x, 1957.
Sabine, C. L., Feely, R. A., Gruber, N., Key, R. M., Lee, K., Bullister, J. L., Wanninkhof, R.,
Wong, C. S., Wallace, D. W. R., Tilbrook, B., Millero, F. J., Peng, T.-H., Kozyr, A., Ono, T.,
and Rios, A. F.: The Oceanic Sink for Anthropogenic $CO_2$, Science, 305, 367-371,
10.1126/science.1097403, 2004.
Sillén, L. G.: Regulation of $O_2$, $N_2$ and $CO_2$ in the atmosphere; thoughts of a laboratory
chemist, Tellus, 18, 198-206, 10.1111/j.2153-3490.1966.tb00226.x, 1966.
Stohl, A., Forster, C., Frank, A., Seibert, P., and Wotawa, G.: Technical note: The Lagrangian
particle dispersion model FLEXPART version 6.2, Atmos. Chem. Phys., 5, 2461-2474,
10.5194/acp-5-2461-2005, 2005.
Takahashi, T., Sutherland, S. C., Wanninkhof, R., Sweeney, C., Feely, R. A., Chipman, D.
W., Hales, B., Friederich, G., Chavez, F., Sabine, C., Watson, A., Bakker, D. C. E., Schuster,
U., Metzl, N., Yoshikawa-Inoue, H., Ishii, M., Midorikawa, T., Nojiri, Y., Körtzinger, A.,
Steinhoff, T., Hoppema, M., Olafsson, J., Arnarson, T. S., Tilbrook, B., Johannessen, T.,
Olsen, A., Bellerby, R., Wong, C. S., Delille, B., Bates, N. R., and de Baar, H. J. W.:
Climatological mean and decadal change in surface ocean $pCO_2$, and net sea–air $CO_2$ flux
over the global oceans, Deep Sea Research Part II: Topical Studies in Oceanography, 56, 554-
577, http://dx.doi.org/10.1016/j.dsr2.2008.12.009, 2009.
NOAA  Earth  System  Research  Laboratory,  Global  Monitoring  Division:
http://www.esrl.noaa.gov/gmd/ccgg/trends/, access: 30.10.2014, 2014.
Schibig, M. F., Steinbacher, M., Buchmann, B., van der Laan-Luijkx, I. T., van der Laan, S.,
Ranjan, S., and Leuenberger, M. C.: Comparison of continuous in situ $CO_2$ observations at
Jungfraujoch using two different measurement techniques, Atmos. Meas. Tech., 8, 57-68,
10.5194/amt-8-57-2015, 2015.



Tans, P. P., Fung, I. Y., and Takahashi, T.: Observational Contrains on the Global
Atmospheric $CO_2$ Budget, Science, 247, 1431-1438, 10.1126/science.247.4949.1431, 1990.
Thompson, D. R., Chris Benner, D., Brown, L. R., Crisp, D., Malathy Devi, V., Jiang, Y.,
Natraj, V., Oyafuso, F., Sung, K., Wunch, D., Castaño, R., and Miller, C. E.: Atmospheric
validation of high accuracy $CO_2$ absorption coefficients for the OCO-2 mission, Journal of
Quantitative       Spectroscopy       and       Radiative       Transfer,       113,       2265-2276,
http://dx.doi.org/10.1016/j.jqsrt.2012.05.021, 2012.
Thoning, K. W., Tans, P. P., and Komhyr, W. D.: Atmospheric carbon dioxide at Mauna Loa
Observatory: 2. Analysis of the NOAA GMCC data, 1974–1985, Journal of Geophysical
Research: Atmospheres, 94, 8549-8565, 10.1029/JD094iD06p08549, 1989.
Trolier, M., White, J. W. C., Tans, P. P., Masarie, K. A., and Gemery, P. A.: Monitoring the
isotopic composition of atmospheric $CO_2$: Measurements from the NOAA Global Air
Sampling Network, Journal of Geophysical Research: Atmospheres, 101, 25897-25916,
10.1029/96JD02363, 1996.
Uglietti, C., Leuenberger, M., and Brunner, D.: European source and sink areas of $CO_2$
retrieved from Lagrangian transport model interpretation of combined $O_2$ and $CO_2$
measurements at the high alpine research station Jungfraujoch, Atmos. Chem. Phys., 11,
8017-8036, 10.5194/acp-11-8017-2011, 2011.
van der Laan-Luijkx, I. T., van der Laan, S., Uglietti, C., Schibig, M. F., Neubert, R. E. M.,
Meijer, H. A. J., Brand, W. A., Jordan, A., Richter, J. M., Rothe, M., and Leuenberger, M. C.:
Atmospheric $CO_2$, $\delta(O_2/N_2)$ and $\delta^{13}CO_2$ measurements at Jungfraujoch, Switzerland: results
from a flask sampling intercomparison program, Atmos. Meas. Tech., 6, 1805-1815,
10.5194/amt-6-1805-2013, 2013.
Vigouroux, C., Blumenstock, T., Coffey, M., Errera, Q., García, O., Jones, N. B., Hannigan,
J. W., Hase, F., Liley, B., Mahieu, E., Mellqvist, J., Notholt, J., Palm, M., Persson, G.,
Schneider, M., Servais, C., Smale, D., Thölix, L. and De Mazière, M.: Trends of ozone total
columns and vertical distribution from FTIR observations at eight NDACC stations around
the globe, Atmospheric Chemistry and Physics, 15(6), 2915–2933, doi:10.5194/acp-15-2915-

29  2015, 2015.

Wunch, D., Toon, G. C., Blavier, J.-F. L., Washenfelder, R. A., Notholt, J., Connor, B. J.,
Griffith, D. W. T., Sherlock, V., and Wennberg, P. O.: The Total Carbon Column Observing





Network, Philosophical Transactions of the Royal Society of London A: Mathematical,
Physical and Engineering Sciences, 369, 2087-2112, 10.1098/rsta.2010.0240, 2011.
Yokota, T., Yoshida, Y., Eguchi, N., Ota, Y., Tanaka, T., Watanabe, H., and Maksyutov, S.:
Global Concentrations of $CO_2$ and $CH_4$ Retrieved from GOSAT: First Preliminary Results,
SOLA, 5, 160-163, 10.2151/sola.2009-041, 2009.
Zander, R., Mahieu, E., Demoulin, P., Duchatelet, P., Roland, G., Servais, C., Mazière, M. D.,
Reimann, S., and Rinsland, C. P.: Our changing atmosphere: Evidence based on long-term
infrared solar observations at the Jungfraujoch since 1950, Science of The Total Environment,
391, 184-195, http://dx.doi.org/10.1016/j.scitotenv.2007.10.018, 2008.
Zellweger, C., Ammann, M., Buchmann, B., Hofer, P., Lugauer, M., Rüttimann, R., Streit, N.,
Weingartner, E., and Baltensperger, U.: Summertime NO y speciation at the Jungfraujoch,
3580 m above sea level, Switzerland, Journal of Geophysical Research: Atmospheres, 105,
6655-6667, 10.1029/1999JD901126, 2000.
Zellweger, C., Forrer, J., Hofer, P., Nyeki, S., Schwarzenbach, B., Weingartner, E., Ammann,
M., and Baltensperger, U.: Partitioning of reactive nitrogen (NOy) and dependence on
meteorological conditions in the lower free troposphere, Atmos. Chem. Phys., 3, 779-796,
10.5194/acp-3-779-2003, 2003.
Zellweger, C., Hüglin, C., Klausen, J., Steinbacher, M., Vollmer, M., and Buchmann, B.:
Inter-comparison of four different carbon monoxide measurement techniques and evaluation
of the long-term carbon monoxide time series of Jungfraujoch, Atmos. Chem. Phys., 9, 3491-
3503, 10.5194/acp-9-3491-2009, 2009.
Zeng, N., Zhao, F., Collatz, G. J., Kalnay, E., Salawitch, R. J., West, T. O., and Guanter, L.:
Agricultural Green Revolution as a driver of increasing atmospheric $CO_2$ seasonal amplitude,
Nature, 515, 394-397, 10.1038/nature13893, 2014.





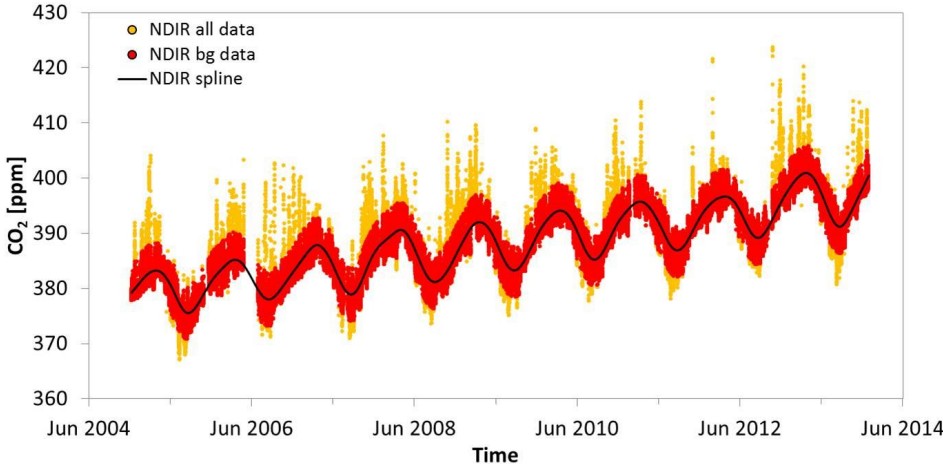

Figure 1. In-situ $CO_2$ mole fractions of the NDIR measurements as a function of time in ppm
at JFJ: All hourly averages before filtering (yellow), hourly averages after filtering (red) and
the spline (black line). Note that the yellow points correspond to only about 5 % of the whole
dataset.





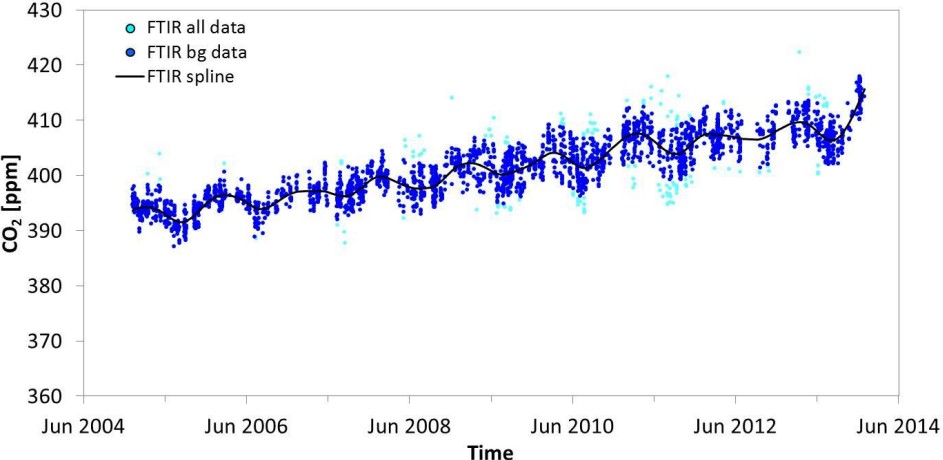

Figure 2. CO$_2$ mole fractions of the FTIR measurements as a function of time in ppm in the
column above JFJ: All hourly averages before filtering (light blue), hourly averages after
filtering (dark blue) and the spline (black line). The light blue points correspond to about 5 %
of the whole dataset.





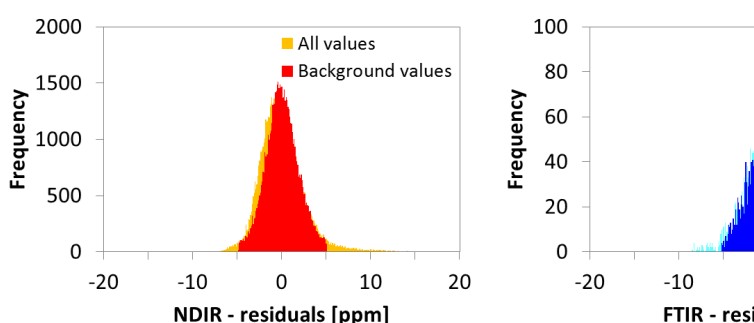

Figure 3. A: Histogram of the filtered NDIR residuals representing the background values
(red) of the in-situ measurements and the rejected values (black); B: Histogram of the filtered
FTIR residuals representing the background values (blue) of the column measurements and
the rejected values (black).





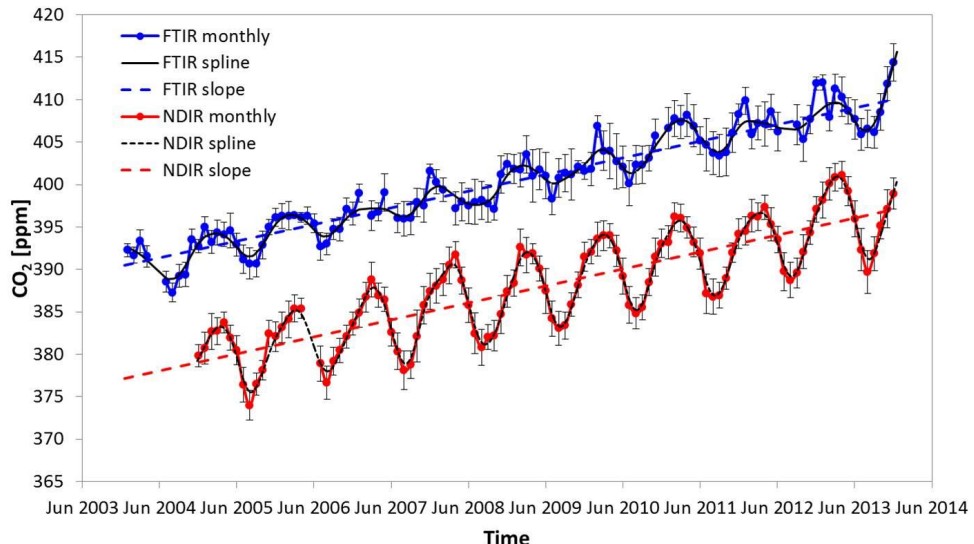

Figure 4. FTIR and NDIR $CO_2$ measurements at JFJ as a function of time: Monthly averages
of the filtered FTIR data (blue), spline (black line), the annual $CO_2$ increase calculated from
the filtered FTIR dataset (blue dashed line), monthly averages of the filtered NDIR data (red),
spline (black dotted line) and the annual $CO_2$ increase calculated from the filtered NDIR
dataset (red dashed line).



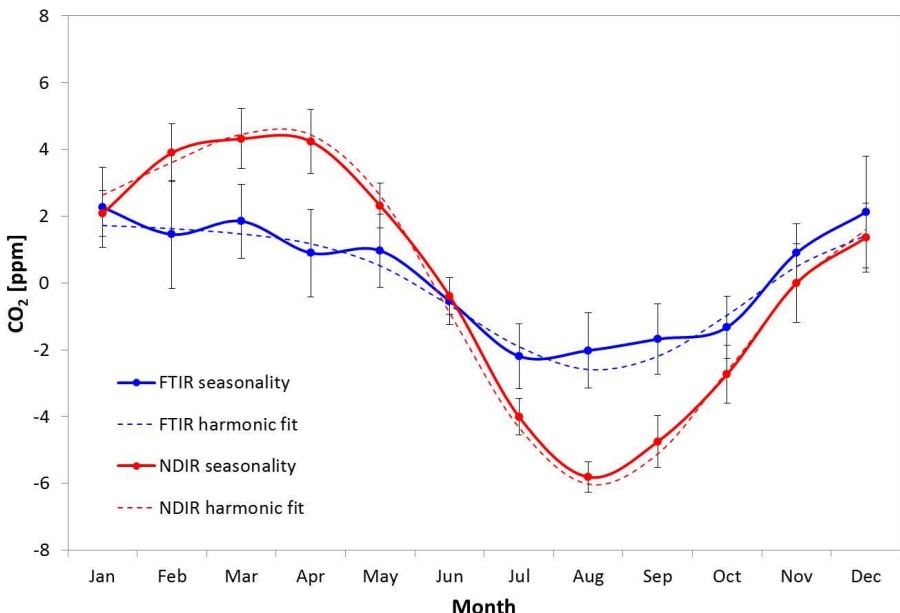

Figure 5. Monthly averaged seasonality of the filtered FTIR and NDIR $CO_2$ measurements for
the nine years of the comparison: averaged NDIR seasonality (red), two harmonic fit of the
NDIR seasonality (red dashed line), averaged FTIR seasonality (blue) and two harmonic fit of
the FTIR seasonality (dashed blue line).




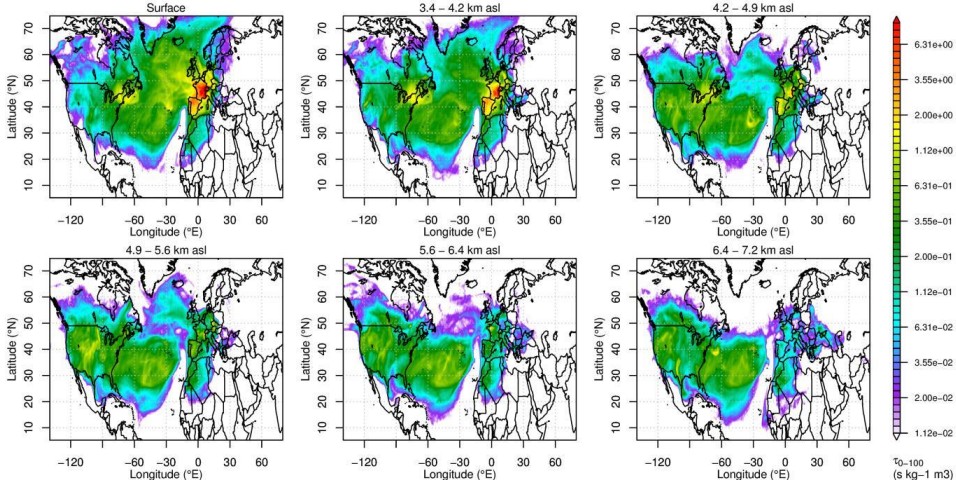

Figure 6. Surface source sensitivity (footprints) of the air masses at JFJ (surface in-situ) and in the sub-columns above JFJ in August ($CO_2$ minimum of FTIR and NDIR time series) in the period 2009 to 2011 simulated with FLEXPART. The height of the sub-columns is given above the according subplots, the x-axis is the longitude, the y-axis represents the latitude, the color code of the sensitivity is given at the right side.





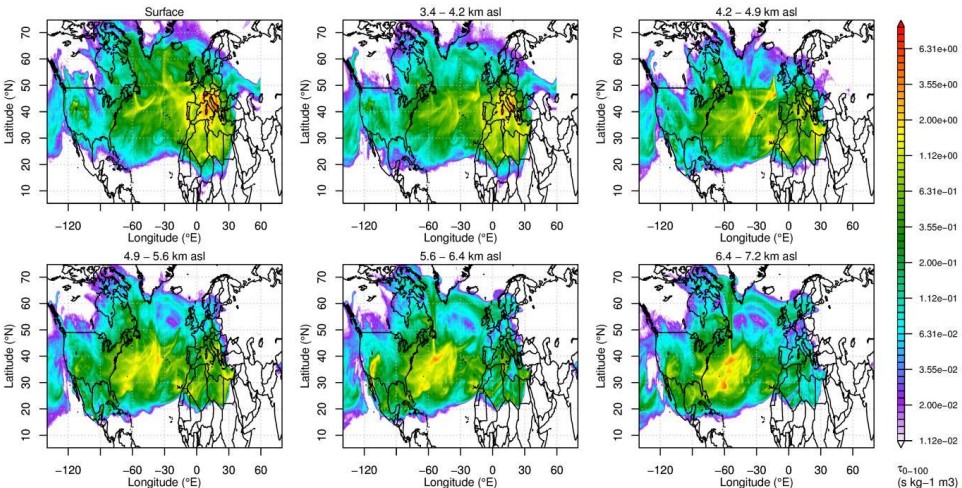

Figure 7. Surface source sensitivity (footprints) of the air masses at JFJ (surface in-situ) and in
the sub-columns above JFJ in January ($CO_2$ maximum of the FTIR dataset) in the period 2009
to 2011 simulated with FLEXPART. The height of the sub-columns is given above the
according subplots, the x-axis is the longitude, the y-axis represents the latitude, the color
code of the sensitivity is given at the right side.





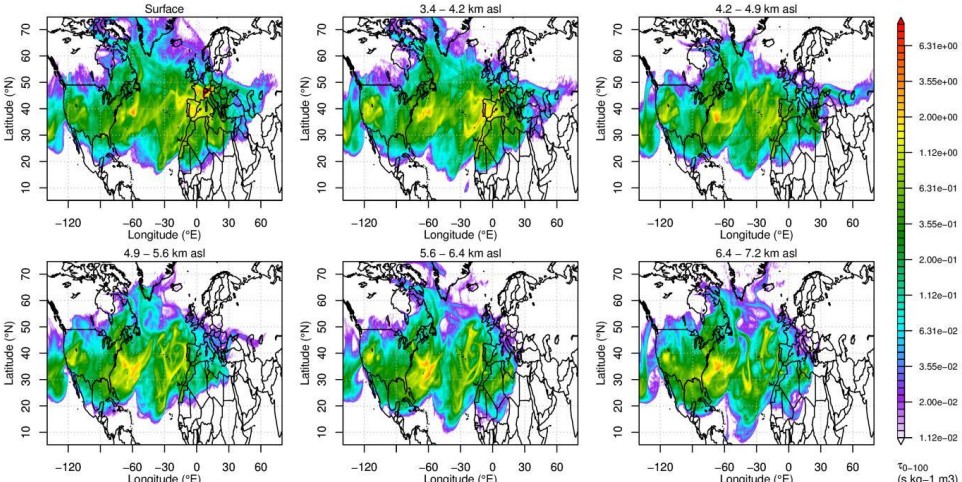

Figure 8. Surface source sensitivity (footprints) of the air masses at JFJ (surface in-situ) and in the sub-columns above JFJ in March ($CO_2$ maximum of the NDIR dataset) in the period 2009 to 2011 simulated with FLEXPART. The height of the sub-columns is given above the according subplots, the x-axis is the longitude, the y-axis represents the latitude, the color code of the sensitivity is given at the right side.



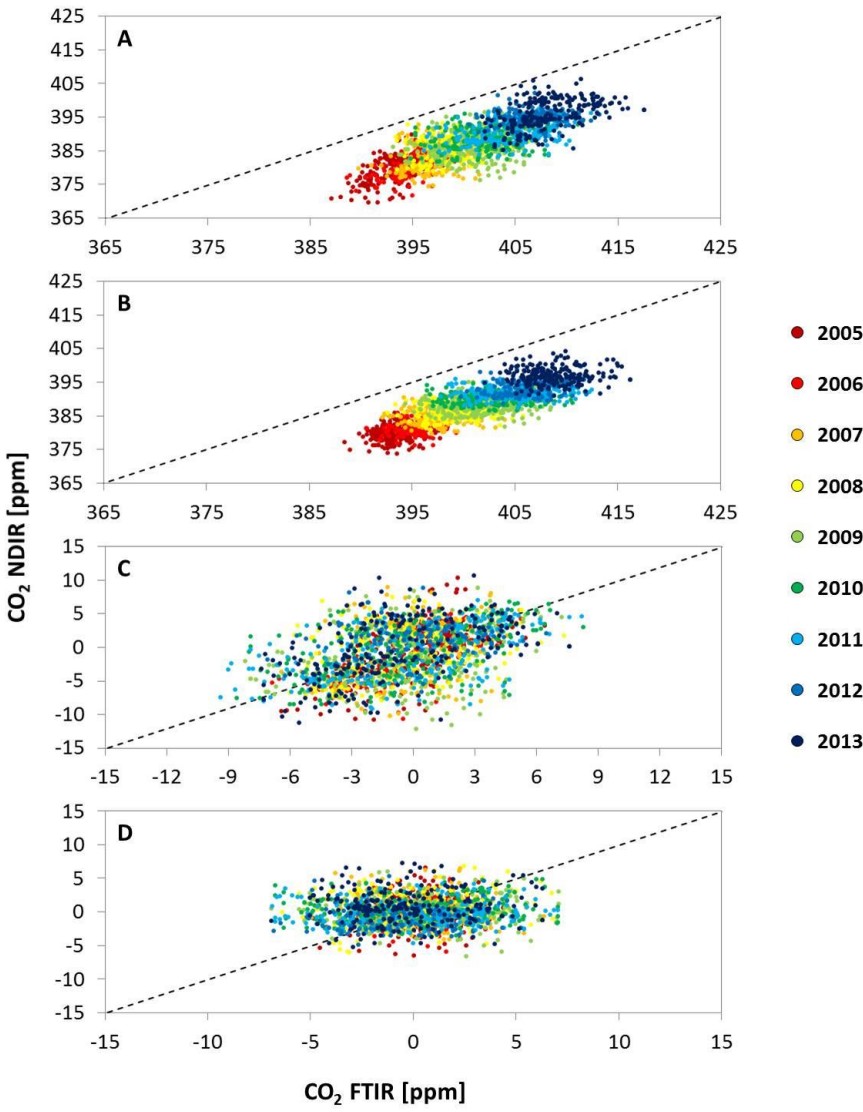

Figure 9. Correlation plots of the filtered hourly NDIR $CO_2$ measurements vs. the filtered

FTIR $CO_2$ measurements. The different colors refer to the years 2005 to 2013 (see legend). A:

The NDIR $CO_2$ measurements vs. FTIR $CO_2$ measurements including both, the annual $CO_2$

increase and the seasonality; B: As A but without seasonality; C: As A but detrended; D: As

A but with neither annual $CO_2$ increase nor seasonality. The dashed line is the 1:1 line.





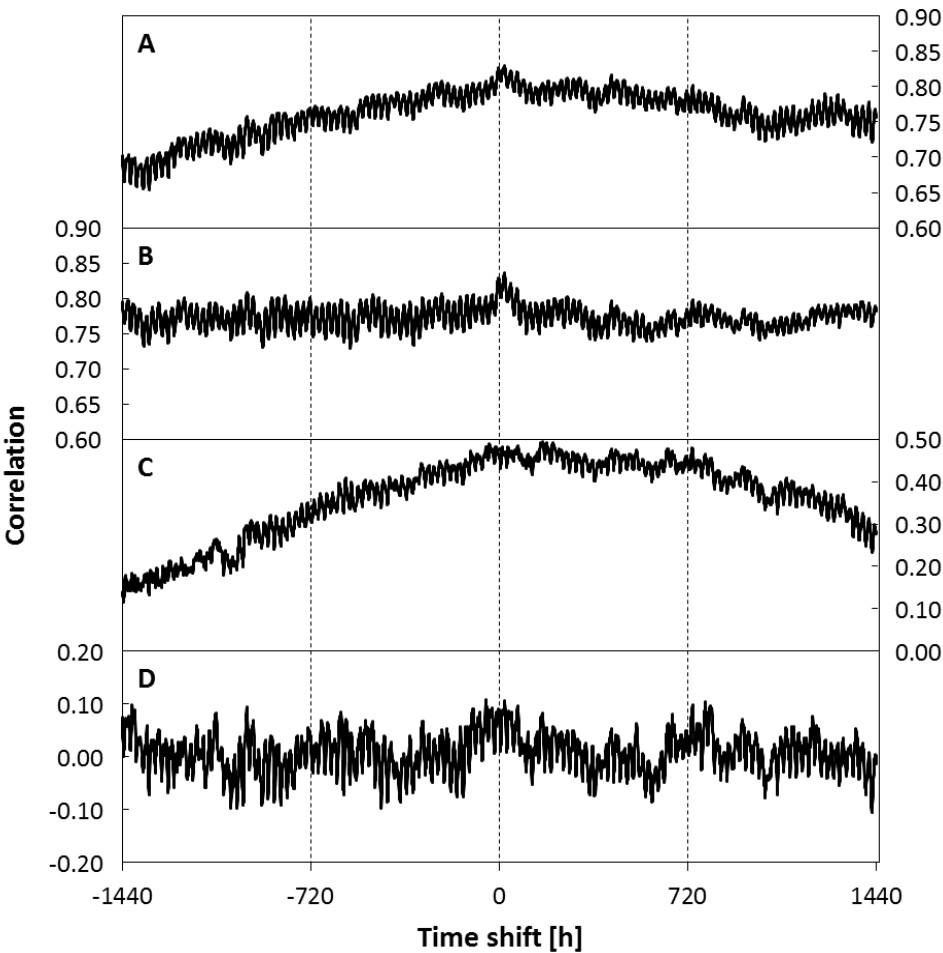

Figure 10. Evolution of the correlation between the filtered FTIR and NDIR datasets with
changing time shift. A: Correlation between complete datasets; B: Correlation between the
two datasets without seasonality; C: Correlation between the two datasets without trend; D:
Correlation between the two datasets with neither trend nor seasonality.



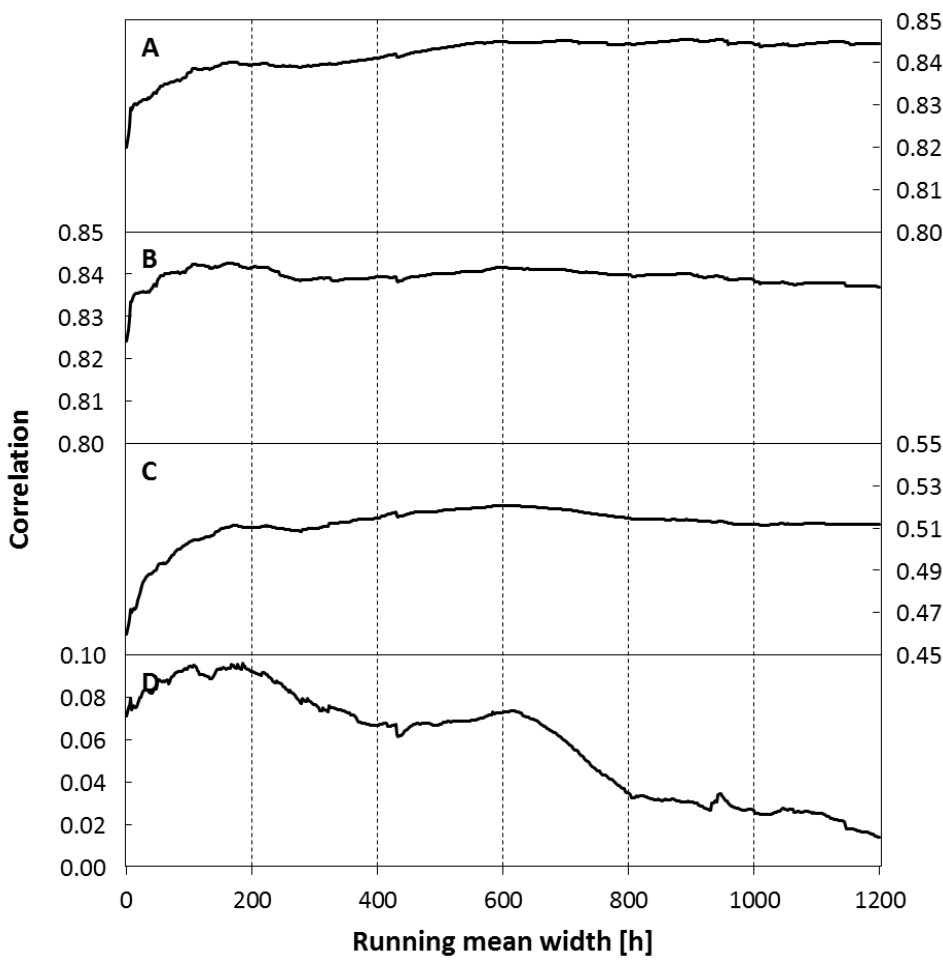

Figure 11. Change of the correlation between the filtered FTIR and NDIR datasets with
increasing width of the running mean. A: Correlation between the two datasets with
seasonality and slope; B: Correlation between the two datasets without seasonality; C:
Correlation between the two datasets without slope; D: Correlation between the two datasets
with neither slope nor seasonality.



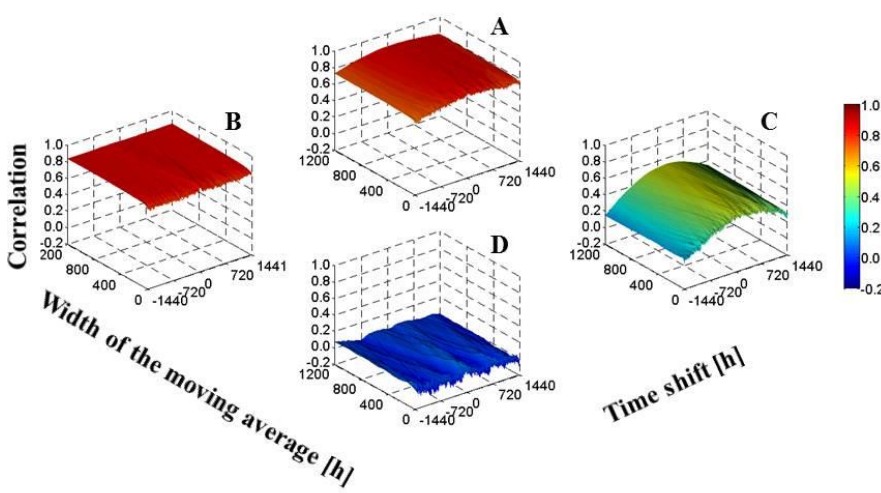

Figure 12. Surface plots of the correlation of the NDIR $CO_2$ measurements vs. the FTIR $CO_2$ measurements. The x-axis corresponds to the time shift, the y-axis to the width of the moving average and the z-axis to the correlation between the FTIR and the NDIR dataset, the color code illustrates the correlation and corresponds to the z-axis values. A: The FTIR $CO_2$ measurements vs. the corresponding NDIR $CO_2$ measurements including the annual $CO_2$ increase as well as the seasonality; B: As A but without seasonality; C: As A but detrended; D: As A but detrended and deseasonalized.