# Peer review of "Intercomparison of in-situ NDIR and column FTIR measurements of CO2 at Jungfraujoch"

_Atmospheric Chemistry and Physics, 2016_

## Referee Comment (RC1) · Anonymous Referee #1 · 4 May 2016

While CO2 is the most important greenhouse gas, its sources and sinks are still not well understood. Studies of the carbon cycle require qualitatively good and long-term measurements. Beside in-situ observations remote sensing observations have become an important tool to study the carbon cycle.

This paper forms an important contribution for such studies. While so far most remote sensing observations are performed in the near-infrared spectral region, organize in TCCON, this paper presents observations in the mid-infrared, organized in the NDACC. Since most NDACC observations cover a longer-time span, it makes sense to perform such studies also in the mid-infrared. This holds especially for the Jungfrauchjoch site, where, together with the Kitt-Peak studies in the US, the longest mid-infrared observations exist. The long-term data set presented, and especially the studies of the seasonality together with the footprint analysis are important and new scientific

contributions.

The paper is well written and I have only a few comments.

Major comments:

The results of the paper depend on the comparability of near-infrared with mid-infrared observations. This needs to be studied in much more detail. Great care has to be taken in order to consider the different sensitivities of both infrared techniques to understand differences and potential biases. Recently two papers have been published where these differences are studied in detail. Barthlott et al, AMT, 2015 and Buschmann et al., AMT, 2016. The authors mention shortly the paper by Barthlott, but do not mention the paper by Buschmann et al. Since the study of the comparability of the mid-infrared data set from Jungfraujoch with near-infrared observations, as performed within TC-CON, are extremely important, the results should be discussed and interpreted with respect to both papers. Besides the presentation of the CO2-data, section 2.3 of the manuscript form the most important part of the paper, and much more details on the analysis should be given.

Minor comments:

The introduction is quite interesting and detailed, but very long. To me, many details about the carbon cycle are not worth mentioning here, this part should be shortened.

For me the Figures 10, 11 and 12 do not tell important new findings. I suggest skipping these Figures, or showing only one instead.

Page 7, line 1: The seasonality is also influenced by fossil fuel combustion, not only by respiration and photosynthesis.

---

## Referee Comment (RC2) · Anonymous Referee #2 · 10 May 2016

Review of 'Intercomparison of in-situ NDIR and column FTIR measurements of CO2 at Jungfraujoch' by Schibig et al.

The paper by Schibig et al., shows a comparison of two very different measurement time series at Jungfraujoch station in Switzerland. Ground-based as well as FTIR column measurements from 2005-2013 are presented. The authors report a consistent trend for both data sets which are in agreement with other stations on the northern hemisphere. The FTIR data set is biased low by 13 ppmv since the stratospheric column reduces the mean column value. The data are filtered for pollution events and clear sky conditions and evidence is provided, that the variability of both data sets is partly due to local CO2 variations. The seasonality is shows very interesting differences between both data sets , which are not explained fully. Both data sets show the seasonal minimum at the same time in August, but different times for the

maximum, which occurs in January for the FTIR data set and in March for the NDIR in-situ measurements. This is explained by different source regions for the respective months on the basis of FLEXPART footprint calculations for 2009-2011. Differences in the vertical distribution are mentioned and particularly the role of the $CO_2$ gradient at the tropopause is not really discussed. In general the manuscript is well written and should be published in ACP, but the analysis of the seasonal differences and the footprint analysis should be sharpened.

Main comments: The NDIR shows the minima in August as well as the FTIR, but the maxima show differences in their time of occurrence. The FTIR shows the maximum for January whereas the NDIR exhibits its maximum in March. I'm not sure if the FLEX-PART footprints in Figs. 6-8 do provide meaningful results for the free tropospheric partial columns. I don't see for a long-lived tracer like $CO_2$ any reason why a ten day backward footprint for the free troposphere should provide an indication of sources and sinks. For the lowest layer this might be valid, but how does the respective footprint explain the seasonal differences in the free troposphere? How different are the footprint distributions in January , March and August from the other months? The different time of occurrence of the respective winter maxima is also not explained by the footprints. Is it maybe caused by seasonality of e.g. warm conveyor belts and therefore seasonality of the vertical tropospheric column? I suggest to analyze the FLEXPART output for this. Which role plays the seasonality of different tropopause height occurrence frequency over JFJ in winter and summer for the interpretation of the $CO_2$ columns and the summer - winter difference between FTIR and NDIR? Further as mentioned in the manuscript also the seasonality in the UTLS modifies the column. Is it possible to quantify this a bit more?

p.5. l. 13: Please specify the long-term stability (i.e. error due to drift) and the total uncertainty of the NDIR.

Technical: Fig.3: The caption refers to black lines or dots, which I can't find. Please correct.

---

## Author Comment (AC1) · 20 Jun 2016

The authors like to thank Referee#1 for his comments and suggestions. The Referee's comments and questions are bold, the authors' replies are formatted as plain text, and excerpts from the manuscript as well as changes to the manuscript are given in italics.

**Reply to Anonymous Referee #1**

**While $CO_2$ is the most important greenhouse gas, its sources and sinks are still not well understood. Studies of the carbon cycle require qualitatively good and long-term measurements. Beside in-situ observations remote sensing observations have become an important tool to study the carbon cycle. This paper forms an important contribution for such studies. While so far most remote sensing observations are performed in the near-infrared spectral region, organize in TCCON, this paper presents observations in the mid-infrared, organized in the NDACC. Since most NDACC observations cover a longer-time span, it makes sense to perform such studies also in the mid-infrared. This holds especially for the Jungfraujoch site, where, together with the Kitt-Peak studies in the US, the longest mid-infrared observations exist. The long-term data set presented, and especially the studies of the seasonality together with the footprint analysis are important and new scientific contributions.**

**The paper is well written and I have only a few comments.**

**Major comments:**

**The results of the paper depend on the comparability of near-infrared with mid-infrared observations. This needs to be studied in much more detail. Great care has to be taken in order to consider the different sensitivities of both infrared techniques to understand differences and potential biases.**

The goal of this study is to compare in-situ measurements of a NonDispersive InfraRed analyzer (NDIR) with column measurements from a Fourier Transform InfraRed (FTIR, mid infrared) system, and to find out whether their different samples (surface air vs. column above the station) provide similar results for the annual $CO_2$ change, the seasonality or even shorter-lived $CO_2$ signals, or not. The physical characteristics of the $CO_2$ adsorption of both methods is not subject of the present study and would be beyond the scope of this publication.

**Recently two papers have been published where these differences are studied in detail. Barthlott et al, AMT, 2015 and Buschmann et al., AMT, 2016. The authors mention shortly the paper by Barthlott, but do not mention the paper by Buschmann et al.**

Buschman et al., (2016) was added to the references and page 9, line 7 was changed from:

*"With a Monte Carlo algorithm, the values of the annual change of the $CO_2$ mole fraction of the two datasets were calculated. Despite the shift between the two datasets of roughly 13 ppm (i.e. about 3%, in line with the systematic uncertainty affecting the FTIR measurement; see section 2.3) and the different measurement techniques the annual $CO_2$ increase is quite similar. The FTIR slope is $2.04 \pm 0.07$ ppm $yr^{-1}$ and the NDIR dataset shows a slope of $1.97 \pm 0.05$ ppm $yr^{-1}$, so they are equal within their uncertainties (Figure 4)."*

to:

*"With a Monte Carlo algorithm, the values of the annual change of the $CO_2$ mole fraction of the two datasets were calculated. Despite the shift between the two datasets of roughly 13 ppm and the different measurement techniques the annual $CO_2$ increase is quite similar. The FTIR slope is 2.04 ± 0.07 ppm yr$^{-1}$ and the NDIR dataset shows a slope of 1.97 ± 0.05 ppm yr$^{-1}$, so they are equal within their uncertainties (Figure 4). The observed offset between the FTIR (NDACC) and in-situ records at Jungfraujoch contrasts the comparison of NDACC and TCCON records as determined at Ny-Ålesund which do not show any offset at all when using several individual $CO_2$ lines for the mid-IR (Buschmann et al., 2016). However, the FTIR/NDIR offset of about 3% is commensurate with the systematic uncertainty affecting the FTIR measurement; see section 2.3."*

**Since the study of the comparability of the mid-infrared data set from Jungfraujoch with near-infrared observations, as performed within TCCON, are extremely important, the results should be discussed and interpreted with respect to both papers.**

Unfortunately, there are no TCCON measurements at Jungfraujoch to compare with.

**Besides the presentation of the $CO_2$-data, section 2.3 of the manuscript form the most important part of the paper, and much more details on the analysis should be given.**

In our opinion, the two measurement systems are equally important. Both measurement systems provide an independent, valuable data set, which we compared.

We added some more detail to the section 2.3. It was changed at page 5, line 3 from:

*"The uncertainty on the main $CO_2$ line strength is estimated at 2 to less than 5% in the HITRAN compilation (Rothman et al., 2005), leading to a systematic error on the retrieved total column of the same magnitude. In the meantime, the data set has been consistently updated, still using the SFIT-1 algorithm (version 1.09c) and a single microwindow spanning the 2024.3 – 2024.7 cm$^{-1}$ spectral interval, whose main spectral line is coming from $^{13}CO_2$. The single $CO_2$ a priori vertical distribution used in all retrievals is characterized by a constant mixing ratio of 338 ppm from the surface up to the tropopause, then slightly decreasing to stabilize at 330 ppm at 20 km and above. A simple scaling retrieval is performed, and the mixing ratio derived for the troposphere is used in the present comparisons. Note that the representativeness of this unique profile is not optimal for all seasons and may lead to an underestimation of the seasonal amplitude (see Fig. 1 in Barthlott et al., 2015), because of a non-optimum vertical sensitivity of the FTIR retrieval. Indeed, typical values of the total column averaging kernel – indicative of the fraction of information coming from retrieval rather than from the a priori (e.g. Vigouroux et al., 2015) – are in the 0.5 – 1 range between the ground and 10 km altitude, in line with Fig. 4 of Barthlott et al. (2015)."*

to:

*"In the meantime, the data set has been consistently updated, still using the SFIT-1 algorithm (version 1.09c) and a single microwindow spanning the 2024.3 – 2024.7 cm$^{-1}$ spectral interval, whose main spectral line at 2024.564 cm$^{-1}$ is coming from $^{13}CO_2$. The uncertainty range on the strength of this $CO_2$ line is estimated at 2 to less than 5 % in the HITRAN compilation (Rothman*

*et al., 2005), leading to a systematic error on the retrieved total column of the same magnitude. The single $CO_2$ a priori vertical distribution used in all retrievals is characterized by a constant mixing ratio of 338 ppm from the surface up to the tropopause, then slightly decreasing to stabilize at 330 ppm at 20 km and above. During the retrieval process, a simple scaling of the whole vertical profile is performed, accounting for interferences by weak ozone and water vapor lines, and the mixing ratio derived for $CO_2$ in the troposphere is used in the present comparisons. Note that the representativeness of this unique profile is not optimal for all seasons and may lead to an underestimation of the seasonal amplitude (see Fig. 1 in Barthlott et al., 2015), because of a non-optimum vertical sensitivity of the FTIR retrieval. Indeed, typical values of the total column averaging kernel – indicative of the fraction of information coming from retrieval rather than from the a priori (e.g. Vigouroux et al., 2015) – are in the 0.5 – 1 range between the ground and 10 km altitude, in line with Fig. 4 of Barthlott et al. (2015). Over all the standard deviation of multiple measurements over the course of a single day corresponds to less than one ppm, which is significantly smaller than the observed seasonal cycle."*

**Minor comments:**

**The introduction is quite interesting and detailed, but very long. To me, many details about the carbon cycle are not worth mentioning here, this part should be shortened.**

The part about the carbon cycle in the introduction was shortened, it reads now from page 2, line 16 to page 3, line 4:

*"$CO_2$ is the most important anthropogenic greenhouse gas, with a large contribution to the greenhouse effect (Arrhenius, 1896) and an additional radiative forcing of the atmosphere currently evaluated at 1.68 Wm-2 (IPCC, 2013). The strength of the forcing is depending on its atmospheric mole fraction which is ruled by the processes of the carbon cycle as well as by anthropogenic $CO_2$ emissions from fossil fuel combustion and land use change. The major reservoirs of the carbon cycle besides the lithosphere are the soils, the ocean, the biosphere and the atmosphere, where the latter is also acting as the main link between the biosphere and the ocean. The linking process between the atmosphere and the ocean is dissolution of $CO_2$ in oceanic water, where it is subsequently chemically bound to bicarbonate and carbonate and therefore removed from the carbon cycle on a longer timescale (Broecker and Peng, 1982;Feely et al., 2004;Heinze et al., 1991;Sillén, 1966). The processes coupling the biosphere with the atmosphere are photosynthesis, where $CO_2$ is taken up by plants, and respiration, where $CO_2$ is released back to the atmosphere. Photosynthesis and respiration are mainly driven by climatic conditions of the environment. In the northern hemisphere, especially in the extratropics with distinct seasons, the dominating process in late spring, summer and fall is photosynthesis and thereby the uptake of $CO_2$ from the atmosphere. In autumn respiration and with it the release of $CO_2$ from the biosphere into the atmosphere starts to take over and is the ruling process in winter until spring when photosynthesis becomes the dominating process again. Due to these alternating processes, the $CO_2$ mole fraction in the atmosphere shows a seasonal cycle with its maximum generally in early spring and its minimum in fall (Halloran, 2012;Keeling et al., 1976;Keeling et al., 2001;Machida et al., 2002)."*

**For me the Figures 10, 11 and 12 do not tell important new findings. I suggest skipping these Figures, or showing only one instead.**

We disagree because if there were significant changes of the correlations with increasing time shifts, increasing widths of the running mean or a combination of the two, these figures would be extremely important because they would show it clearly. However, the lack of clear changes in the correlation in combination with the sensitivity plots indicates that the short term variability of the two signals can't be compared. Therefore we like to keep these figures.

**Page 7, line 1: The seasonality is also influenced by fossil fuel combustion, not only by respiration and photosynthesis.**

We changed the sentence:

*"One is the linear increase due to fossil fuel combustion (trend) and one is the annual in- and decrease due to respiration and photosynthesis (seasonality)."*

to:

*"One is the linear increase due to fossil fuel combustion (trend) and one is the annual in- and decrease due to respiration and photosynthesis, and to a lesser degree due to fossil fuel combustion (seasonality)."*

---

## Author Comment (AC2) · 20 Jun 2016

The authors like to thank Referee#2 for his comments and suggestions. The Referee's comments and questions are bold, the authors' replies are formatted as plain text, and excerpts from the manuscript as well as changes to the manuscript are given in italics.

**Reply to Anonymous Referee #2**

**Review of 'Intercomparison of in-situ NDIR and column FTIR measurements of CO2 at Jungfraujoch' by Schibig et al.**

**The paper by Schibig et al., shows a comparison of two very different measurement time series at Jungfraujoch station in Switzerland. Ground-based as well as FTIR column measurements from 2005-2013 are presented. The authors report a consistent trend for both data sets which are in agreement with other stations on the northern hemisphere. The FTIR data set is biased low by 13 ppmv since the stratospheric column reduces the mean column value.**

We would have expected the FTIR dataset to be slightly lower, because of the lower $CO_2$ mole fraction in the stratosphere, but since the FTIR data set is biased high by 13 ppm, we think this is caused by the uncertainty in the HITRAN compilation, which leads to a systematic error on the retrieved total column values. Further, we expect the influence of the stratosphere on the FTIR measurements to be significantly less than 13 ppm.

**The data are filtered for pollution events and clear sky conditions and evidence is provided, that the variability of both data sets is partly due to local $CO_2$ variations. The seasonality is shows very interesting differences between both data sets, which are not explained fully. Both data sets show the seasonal minimum at the same time in August, but different times for the maximum, which occurs in January for the FTIR data set and in March for the NDIR in-situ measurements. This is explained by different source regions for the respective months on the basis of FLEXPART footprint calculations for 2009-2011. Differences in the vertical distribution are mentioned and particularly the role of the $CO_2$ gradient at the tropopause is not really discussed. In general the manuscript is well written and should be published in ACP, but the analysis of the seasonal differences and the footprint analysis should be sharpened.**

**Main comments:**

**The NDIR shows the minima in August as well as the FTIR, but the maxima show differences in their time of occurrence. The FTIR shows the maximum for January whereas the NDIR exhibits its maximum in March. I'm not sure if the FLEXPART footprints in Figs. 6-8 do provide meaningful results for the free tropospheric partial columns. I don't see for a long-lived tracer like $CO_2$ any reason why a ten day backward footprint for the free troposphere should provide an indication of sources and sinks. For the lowest layer this might be valid, but how does the respective footprint explain the seasonal differences in the free troposphere?**

The vertical transport time scale in the troposphere is usually smaller than 10 days (as used in our FLEXPART simulations). Therefore, the model particles are usually widely dispersed in the troposphere after 10 days of transport. Although they won't be well mixing within the whole

northern hemispheric troposphere, the influence of surface source regions beyond the 10 day transport is usually sufficiently diluted and one does not find distinct signals from any specific source region. This is also true for free tropospheric release (receptor) locations since horizontal transport is faster in the troposphere and despite the absence of significant turbulent dispersion the particle plumes disperse due to wind field divergences. Therefore, we are convinced that the 10 day transport scale and derived surface residence times are sufficient to allow a qualitative interpretation of the contribution from different potential source areas.

**How different are the footprint distributions in January, March and August from the other months? The different time of occurrence of the respective winter maxima is also not explained by the footprints. Is it maybe caused by seasonality of e.g. warm conveyor belts and therefore seasonality of the vertical tropospheric column? I suggest to analyze the FLEXPART output for this.**

The footprints for the selected months are fairly representative for the respective season, with the exception of the January footprints which revealed strong influence from northern Africa at different vertical levels, which was not observed in other winter months. In order to further analyze the influence on transport on the observed seasonal cycle, we analyzed the timing of surface influence for different land regions and present this as a new figure and section in the revised manuscript. This extended transport analysis is able to explain the observations in the sense that we find an increased decoupling between the free troposphere and the land surface north of 30°N during the winter months, whereas the influence from tropical land surfaces south of 30°N was increased in winter. Both suggests lower $CO_2$ in the FT (free troposphere) than at the surface and an interruption of the wintertime increase in the FT above JFJ due to the onset of the decoupling and tropical influence just following the observed maximum in February.”

The following section was added at page 9, line 29:

“*In general, the decoupling between the FTIR columns and possible surface fluxes of $CO_2$ from land surfaces north of 30°N was strongest during the winter month (January to March), when especially low surface residence times were simulated by FLEXPART for the free tropospheric FTIR columns (Figure 9). From April to September larger surface residence times were seen also for the FTIR columns and a stronger coupling between surface fluxes and the free troposphere can be expected. At the same time residence times over tropical land surface (south of 30°N) were generally larger for the FTIR columns and were especially increased from February to April (see Figure 9).*

*and page 13, line 6:*

“*...2009). The findings based on Figure 9 can help to understand the shift in the observed wintertime maximum of $CO_2$ between FTIR (January) and NDIR (March-April) The land surfaces of northern hemispheric mid-latitudes act as a net $CO_2$ source during the winter half year, since photosynthesis is largely reduced and respiration and anthropogenic emissions of $CO_2$ dominate the budget, hence, the observation of maximum $CO_2$ at the end of the winter half year and close to the surface. For the free troposphere above JFJ as observed by the FTIR the direct link to these wintertime releases of $CO_2$ is weakened due to generally reduced vertical transport. At the same time more frequent transport from and land surface contact in the tropics can be deduced, an area that even during the winter half year may act as a net $CO_2$ sink due to photosynthetic*

*uptake. An earlier onset of decreasing $CO_2$ in the FT above Jungfraujoch could thereby be explained by different seasonality of transport and vertical mixing. Additionally… "*

And the following figure with caption was added as Figure 9:

[Figure]

*Figure 9, Annual cycle of FLEXPART derived total surface residence time over land for different vertical arrival columns above Jungfraujoch: (left) for land surfaces north of 30°N and (right) for land surfaces south of 30°N.*

**Which role plays the seasonality of different tropopause height occurrence frequency over JFJ in winter and summer for the interpretation of the $CO_2$ columns and the summer - winter difference between FTIR and NDIR? Further as mentioned in the manuscript also the seasonality in the UTLS modifies the column. Is it possible to quantify this a bit more?**

Indeed, this is an interesting, important, and valid point that hasn't been addressed in the present work, therefore we cannot adequately reply to it. Generally, we would expect a lower tropopause could potentially lower the column integrated $CO_2$ value due to the expected lower stratospheric $CO_2$ mole fraction. A detailed analysis regarding this issue requires substantial additional modeling, which was not possible within this work.

**p.5. l. 13: Please specify the long-term stability (i.e. error due to drift) and the total uncertainty of the NDIR.**

The value given in the manuscript corresponds to the standard deviation of several cylinder measurements each lasting at least one hour. The gas from the cylinders was treated, calibrated, and evaluated exactly the same way as outside air, which is why we consider this standard deviation as the precision of our system. The long term stability is taken care of by frequent measurements of calibration gases (see section 2.2).
To make this clearer, we changed the sentence at page 5, line 14 from:

*"Cylinder measurements with a known mole fraction showed a precision better than 0.04 ppm for 1 hour analysis."*

to:

*"Cylinder measurements with a known mole fraction showed a long-term precision for hourly averages better than 0.04 ppm. The accuracy of our target cylinder corresponds to less than 0.1 ppm (WMO target value for CO$_2$ measurements) calculated as standard deviation of the mean considering the number of independent calibration set (high span, low span, working gas)."*

**Technical: Fig.3: The caption refers to black lines or dots, which I can't find. Please correct.**

That's correct, the caption refers to an older version of the figure. It was changed to:

*"Figure 3. A: Histogram of all NDIR residuals (yellow) and the filtered NDIR residuals representing the background values (red) of the in-situ measurements; B: Histogram of all FTIR residuals (light blue) and the filtered FTIR residuals representing the background values (blue) of the column."*

**Changes on the authors' behalf:**

The wavelength of the NDIR analyzer was added, p. 5, line 8 was changed from:

*"...NDIR spectrometer (Maihak S710) with a frequency ..."*

To:

*"...NDIR spectrometer (Maihak S710) measuring at a wavelength of 4.26 μm with a frequency..."*

The Figures' numbers were updated because of the additional figure.

For more clarity, page 15, line 27 was changed from:

*"...or (c) since the FTIR retrievals has little vertical sensitivity the measured column signal contains mixed information from the troposphere and the stratosphere."*

to:

*"...or (c) since the FTIR vertical sensitivity was not exploited in the present retrievals the measured column signal contains mixed information from the troposphere and the stratosphere."*

The reference of Rothman et al., (2005) at page 22, line 19 was moved down after Revelle et al. (1957), to maintain the correct alphabetical order.

---

## Author Response (AR2)

The authors like to thank the editor for his comments and suggestions. The Editor's comments are given in bold, the authors' answers are formatted as plain text and changes to the manuscript are given in italics.

**Editor Decision: Reconsider after minor revisions (Editor review) (22 Jun 2016) by**

**Andreas Engel**

**Comments to the Author:**

**The authors have adressed most issues raised by the reviewers. There are a few points**

**where I suggest some additional clarification:**

**\* The new Figure 9 needs some explanation. Please explain more carefully what the**

**surface residence time is and how it is calculated.**

At page 8, line 23 we added the following

*"...source sensitivities, represented by total residence time, which were calculated as total*

*residence times derived by summing residence times over all start times and over all*

*integration time steps within a selected integration interval for all grid cells as given in*

*Henne et al. (2010)."*

**Reviewer #2 questioned the usefulness of figure 6-8. In general it seems to me that the**

**main message of these two figures is now condensed in Figure 9. I suggest to consider**

**either leaving out figures 6-8 or explaining them a little more. Right now they are**

**presented and very little discussion is actually given to the figures.**

Since figure 9 is not just a summary of figure 6-8 but gives additional information we decided to keep all figures but changed and extended the discussion of figures 6-8 as follows:

*"Sensitivity analyses revealed that the upper tropospheric air originates from different*

*geographic regions, mainly from south west, than the in-situ air measured by the NDIR.*

*During summer, the NDIR measurements record mainly air from European regions, whereas*

*the FTIR sees more influence from the west (Figure 6). From winter to spring, NDIR $CO_2$*

*values are again driven by European sources, whereas FTIR values represent a significantly*

*wider foot print reaching to west and further to the north in contrast to the summer situation*

*(Figure 7, Figure 8). Similar studies investigating CO at JFJ also showed that JFJ is not only*

*sensitive to Central Europe but also to regions as far west as for example North America, the*

*Pacific or even Asia and that the influence of these regions is getting stronger with increasing*

*height (Dils et al., 2011;Pfister et al., 2004;Zellweger et al., 2009). Therefore the air measured by the FTIR is partially decoupled from the increasing $CO_2$ values of the winter-time northern hemisphere. Furthermore, the decoupling might be amplified by the weak overturn of tropospheric air in winter. Towards spring, the tropospheric overturn speeds up again which results in synchronous $CO_2$ minima for both datasets in August (Figure 9).These findings can help to understand the shift in the observed wintertime maximum of $CO_2$ between FTIR (January) and NDIR (March-April). The land surfaces of northern hemispheric midlatitudes act as a net $CO_2$ source during the winter half year, since photosynthesis is largely reduced and respiration and anthropogenic emissions of $CO_2$ dominate the budget. Hence, the maximum of $CO_2$ is observed at the end of the winter half year and close to the surface. For the free troposphere above JFJ as observed by the FTIR the direct link to these wintertime releases of $CO_2$ is weakened due to generally reduced vertical transport. At the same time more frequent transport from and land surface contact in the tropics can be deduced (Figure 9), an area that even during the winter half year may act as a net $CO_2$ sink due to photosynthetic uptake."*

**The answer to the remark of reviewer #2 regarding the seasonality of the UTLS is not satisfactory. I suggest to at least discuss the difference in seasonality between stratosphere and troposphere and point out that this will influence the column but not the ground measurments, procifing a further possible shift in seasonality. I think that the magnitude of the possible effect can be discussed without the need for a large model study.**

We consulted several publications related to this topic namely Nakazawa et al. (1990), Hoor et al., (2004), Ray et al. (2014), Engel et al. (2006), Sawa et al. (2008), Sawa et al. (2015). These publications document a consistent anti-correlation between tropospheric and stratospheric $CO_2$ mole fractions. There are two effects that were put forward to explain this anti-correlation namely (i) the mixing of uplifted tropical air masses horizontally transported to northern latitudes with locally uplifted tropospheric air; (ii) only vertical transport from the troposphere through the tropopause into the lowermost stratosphere resulting in a damped and time lagged stratospheric $CO_2$ signal.

From these publications we would guess that the influence of the tropical air transport to latitudes of our investigations (46°N) has a minor influence on the phasing between tropospheric and stratospheric $CO_2$ evolutions, due to the small phase shift of about 2 months in the $CO_2$ signal between the tropics and the mid-latitudes. Therefore we expect the direct vertical transport through the tropopause to be the main influence for the observed phase shift between the tropospheric and stratospheric $CO_2$. Based on that, we estimate that the FTIR

measurements represent a pressure weighted mean of tropospheric and stratospheric $CO_2$

mole fractions being anti-correlated. A simple calculation assuming the stratospheric signal to be anti-correlated and damped to an amplitude of 30% (4.1 ppm) compared to the in-situ observations at Jungfraujoch leads to an amplitude that corresponds rather nicely with the

FTIR observations, which is 4.4 ppm. The evolution of $CO_2$ mole fraction over the course of a year results in an excellent agreement with the FTIR observations during summer, which seems to be mainly influenced by direct vertical transport. In contrast, during spring there is a substantial mismatch of about 1.5 – 2 ppm.

Since these are very simple calculations we hesitate to include them into the manuscript.

However, we would agree to change the manuscript accordingly if you as editor ask for it. In this case we would suggest a similar text to be included as shown above.

**The change suggested by the author on p. 15. l 27 regarding the non-exploitation of the**

**vertical resolution of the FTIR data is misleading, as it may suggest that the FTIR has a**

**good vertical resolution which was just not exploited. Please merge the old and the new**

**statement, explaining that vertical resolution is poor and was therefore not exploited.**

The sentence was changed from:

*"...or (c) since the FTIR vertical sensitivity was not exploited in the present retrievals the*

*measured column signal contains mixed information from the troposphere and the*

*stratosphere."*

to:

*"...or (c) since the FTIR has a low vertical sensitivity it was not exploited in the present*

*retrievals and therefore the measured column signal contains mixed information from the*

*troposphere and the stratosphere."*

Change on the authors' behalf:

Page 10, line 11 was changed from:

*"… the FTIR columns and were especially increased…"*

To:

*"...the FTIR columns compared to the surface and were especially increased..."*

[revised manuscript text omitted]

---

## Author Response (AR3)

The authors like to thank the editor for his comments and suggestions. The Editor's comments are given in bold, the authors' answers are formatted as plain text and changes to the manuscript are given in italics.

**Co-Editor Decision: Reconsider after minor revisions (Editor review)** (01 Jul 2016) by

Andreas Engel

The authors of this paper like to thank the editor for the time and effort he put in assessing this paper and we hope our answers are satisfying.

We marked the author's comments in bold italics, whereas the author's answers are in a normal typeset. The changes are also marked in yellow in the manuscript following the point- by-point response.

**Comments to the Author:**

**Most remaining questions have been clarified.**

**There are two issues, where I would like to see some improvements.**

**The first one is the description of the residence time. Please be a little more explicit on**

**what this represents. It is then o.k. to refer to Henne et al. for the way it was calculated.**

Page 8 line 22-31 was changed from:

*"Particles released at and above JFJ were followed 10 days backward in time to calculate*

*source sensitivities, represented by total residence time, which were calculated as total*

*residence times derived by summing residence times over all start times and over all*

*integration time steps within a selected integration interval for all grid cells as given in*

*Henne et al. (2010). Source sensitivities were evaluated on regular longitude/latitude grids.*

*The resolution was 0.5° x 0.5° globally, 0.2° x 0.2° over Europe and an even higher*

*resolution of 0.1° x 0.1° was used in the Alpine area. The footprints of the individual*

*measurements of each partial column were averaged to monthly means to get information*

*about the origin of the air masses in the according month (Henne, 2014;Henne et al., 2013)."*

to:

*"Particles released at and above JFJ were followed 10 days backward in time by simulating*

*atmospheric transport by the mean wind, turbulence and convection. Along the integration*

*the particle positions were evaluated every 3 hours to derive particle residence times close to*

*the surface (0 to 100 m above model ground). The residence times give a direct link between*

*concentrations at the receptor (here location of observations) and a source on the evaluated*

*output grid. Hence, residence times are also often termed source sensitivities or concentration*

*footprints. For individual backward simulations total residence times were calculated by*

*summation over all transport integration steps. Larger total residence times usually indicate*

*a larger probability that an air mass was influenced by fluxes at the Earth's surface, whereas*

*lower values indicate air masses that mainly resided in the free troposphere prior to arrival*

*at the receptor. Surface residence times were evaluated on regular longitude/latitude grids.*

*The resolution was 0.5° x 0.5° globally, 0.2° x 0.2° over Europe and an even higher*

*resolution of 0.1° x 0.1° was used in the Alpine area. The surface residence times*

*corresponding to each measurement and each partial column were averaged to monthly*

*means to get information about the origin of the air masses in the according month (Henne,*

*2014;Henne et al., 2013). Further summation over all land cells in the output grid gives an*

*integrating parameter for potential surface influence."*

"Henne, 2014" refers to a technical report within an EU project and we leave it to the editor to decide whether he likes to keep this reference or not.

**The second point is the phase shift with respect to the stratosphere. The reason for the phase shift in the stratosphere is not a shift in the seasonal cycle between the tropics and mid-latitudes in the troposphere. The different phase of the seasonal cycle in the lower stratosphere is caused by a time lag for tropospheric air to reach that area. I suggest to quantitatively mention that the seasonal cycle in the lower stratosphere is shifted with respect to the troposphere, that this will influence the column and the seasonal cycle of the column, but not the ground based measurements and that this should be explored more closely in future studies. Modelling the seasonal cycle of $CO_2$ in the lower stratosphere is far from trivial, so it is plausible that a quantitative assessment of this effect is beyond the scope of this study.**

*At Page 13 line 28 the following was added: "Additionally, the phase of the stratosphere's seasonal cycle is shifted with respect to the tropospheric seasonal cycle because there's time lag for tropospheric air reaching the stratosphere. (Ray et al., 2014, Sawa et al., 2015, Sawa et al., 2008) This effect is only seen by the column measurements of the FTIR system but not by the NDIR system and therefore possibly adds to the differences in the seasonalities of the two data sets."*

and at page 13 line 30: *"To model and quantify these effects properly is rather difficult and*

*beyond the scope of this study, but could be investigated in a following study."*

The references were updated accordingly.

Changes on the author's behalf:

Henne et al. 2015 was published in ACP in the meantime, therefore at page 8 line 13, the reference has to be changed to:

*"(Henne et al, 2016)"*

and accordingly in the references section to:

[revised manuscript text omitted]